

# Sea-ice and water dynamics and moonlight impact the acoustic backscatter diurnal signal over the eastern Beaufort Sea continental slope

Igor A. Dmitrenko[1], Vladislav Petrusevich[1], Gérald Darnis[2], Sergei A. Kirillov[1], Alexander S.
Komarov[3], Jens K. Ehn[1], Alexandre Forest[2], Louis Fortier[2], Søren Rysgaard[1,4] and David G. Barber[1]

[1]Centre for Earth Observation Science, University of Manitoba, Winnipeg, R3T 2N2, Canada
[2]Department of Biology, Laval University, Québec City, G1V 0A6, Canada
[3]Data Assimilation and Satellite Meteorology Research Section, Environment and Climate Change Canada, Ottawa, K1V
1C7, Canada
[4]Arctic Research Centre, Aarhus University, Aarhus, DK-8000, Denmark

*Correspondence to*: Igor A. Dmitrenko (igor.dmitrenko@umanitoba.ca)

**Abstract.** A two-year-long time series of currents and acoustic backscatter from an Acoustic Doppler Current Profiler, moored over the eastern Beaufort Sea continental slope from October 2003 to September 2005, were used to assess dynamics and variability of the sound-scattering layer. It has been shown that acoustic backscatter is dominated by a synchronized diel vertical migration (DVM) of the zooplankton. Our results show that DVM timings (i) were synchronous with sunlight, and (ii) were modified by moonlight and sea-ice, which attenuates light transmission to the water column. Moreover, DVM is modified or completely disrupted during highly energetic current events. Thicker ice observed during winter 2004-2005 lowered the backscatter values, but favored extending DVM toward the midnight sun. In contrast to many previous studies, DVM occurred through the intermediate water layer during the ice-free season of the midnight sun in 2004. In 2005, the midnight sun DVM was likely masked by a high acoustic scattering generated by suspended particles. During full moon at low cloud cover, the nighttime moonlight illuminance led to zooplankton avoidance of the sub-surface layer disrupting DVM. Moreover, DVM was disrupted by upwelling, downwelling and eddy passing. We suggest that these deviations are consistent with DVM adjusting to avoid enhanced water dynamics. For upwelling and downwelling, zooplankton likely respond to the along-slope water dynamics dominated by surface- and depth-intensified flow, respectively. This drives zooplankton to adjust DVM by aggregating in the low or upper intermediate water layer for upwelling and downwelling, respectively. The baroclinic eddy reversed DVM below the eddy core.

## 1 Introduction

The acoustic backscatter signal recorded in the ocean by acoustic Doppler current profilers (ADCPs) is mainly dominated by zooplankton. The diurnal patterns of the acoustic backscatter signal are comprised of diel vertical migration (DVM) of zooplankton, the synchronized movement of zooplankton up and down in the water column over a daily cycle (e.g., Brierley,



2014). In terms of biomass, DVM is arguably the largest daily migration of animals on earth (Hays, 2003), and the largest non-human migration (Brierley et al., 2014). DVM has been extensively explored in the Arctic using either echo sounders or zooplankton nets (e.g., Kosobokova, 1978; Fortier, et al., 2001; Blachowiak-Samolyk et al., 2006; Cottier et al., 2006; Falk-Petersen et al., 2008). The latest progress in assessing DVM in the Arctic is related to understanding DVM during the Arctic

polar night (Berge et al., 2009, 2015) and the role of moonlight in modifying DVM (Last et al., 2016; Petrusevich et al., 2016). While significant progress has been achieved in understanding DVM, the sea-ice and ocean dynamics control on DVM in the Arctic environment remains poorly appreciated.

ADCPs moored over the entire annual cycle in the seasonally ice-covered Arctic water provide a unique temporal evolution of the DVM patterns. This seasonal perspective is essential to achieve a more complete and quantitative

understanding of DVM in response to the light and sea-ice conditions (e.g., Tran et al., 2016; Hobbs et al., 2018). Here we assess temporal evolution of the DVM patterns using a two-year-long time series of velocity and acoustic backscatter from an ADCP-equipped mooring deployed over the upper eastern Beaufort Sea continental slope from October 2003 to September 2005 (Figure 1). The ADCP limitation, however, comes from its ability to detect only the biomass moving at a population level, i.e., comprising the migrating sound-scattering layer (Hobbs et al., 2018). We partly overcame this

disadvantage by using zooplankton samples collected in September 2016 in the southeastern Beaufort Sea (Figure 1a). The environmental factors controlling DVM in the seasonally ice-covered Arctic areas, located at the inner border of the polar circle, remains poorly assessed. Here we use observations from the oceanographic mooring located at ~71°N, the area, where the Sun is between 0 and <6° below the horizon all day on the winter solstice. This latitude corresponds to the Polar twilight when solar illumination is still sufficient for the human eye to distinguish terrestrial objects during short winter

daylight hours. This geographical position makes our DVM observational site vastly different from those at Svalbard (nautical polar night, ~80°N; e.g., Grenvald et al., 2016; Darnis et al., 2017), Canada Basin (civil polar night, ~77.5°N; La et al., 2018), and Northeast Greenland (civil polar night, ~74.5°N, Petrusevich et al., 2016)

This study is built on results by Dmitrenko et al. (2018) on water dynamics over the eastern Beaufort Sea continental slope taking advantage of using the ADCP-derived acoustic backscatter for temporal appreciation of DVM

patterns during two consecutive annual cycles. Our particular focus is on the DVM modifications, caused by wind-forced upwelling and downwelling over the Beaufort Sea continental slope, and the different types of sea-ice cover. We also add more data points and further proof to research focused on the effect of moonlight on DVM (e.g., Webster et al., 2015; Last et al., 2016; Petrusevich et al., 2016).

## 2 Data

We used data from the ArcticNet oceanographic mooring CA13 deployed over the upper Canadian Beaufort Sea continental slope at 300-m depth from 9 October 2003 to 4 September 2005 at 71°21.356'N, 228°38.176'E (Figure 1). The mooring description can be found in Dmitrenko et al. (2016). For this study, we used (i) velocity and acoustic backscatter intensity records from a 300 kHz upward-looking Workhorse Sentinel ADCP by Teledyne RD Instruments (RDI) at 119-m depth and





(ii) temperature records from the moored CTD (temperature-salinity-depth) SBE-37 by Sea-Bird Electronics, Inc. at 49-m
and 119-m depth. The velocity and acoustic backscatter data were obtained at 8-m depth intervals, with a 1-h ensemble time
interval and 30 pings per ensemble. The first bin was located at ~9 m above the transducer, i.e., at 108 m depth. For this
research, we used bins at 28, 68, and 108 m depth. Data at 48 and 88 m depth were obtained by linear interpolation between
bins at 44 m and 52 m, and 84 m and 92 m, respectively. The RDI ADCP precision and resolution are $\pm$ 0.5% and $\pm$ 0.1 cm
s$^{-1}$, respectively. The standard deviation for an ensemble average of 30 pings for the 8-m depth cell size is reported by RDI
to be 1.19 cm s$^{-1}$. The accuracy of the ADCP vertical velocity measurements is not validated; however, the RDI reports that
the vertical velocity is more accurate than the horizontal velocity by at least a factor of two. The compass accuracy is $\pm$ 2°.
The magnetic deviation was added. The along-slope direction was determined to be 64°T (°T - the direction measured with
reference to the true north) using the scatterplot of the daily mean velocity data following in assumption that the maximum
dispersion of velocity measurements occurs along the continental slope (Dmitrenko et al., 2016). Mooring data were
complemented by the vertical CTD, chlorophyll fluorescence and particulate beam attenuation profiles taken at mooring
deployment and recovery in October 2003 and September 2005, respectively, and in July 2004 using a CTD probe SBE-911
(Figure 2). According to the manufacturers' estimates, individual temperature and conductivity measurements are accurate to
$\pm$0.001°C and $\pm$0.0003 S m$^{-1}$, respectively, for the SBE-911, and to $\pm$0.002°C and $\pm$0.0003 S m$^{-1}$ for SBE-37.

The total cloud cover (%) for the mooring location is obtained from the National Centers for Environmental
Prediction - NCEP (Kalnay et al., 1996). The accuracy of the cloud cover data is uncertain. Comparing the satellite- to
NCEP-derived cloud cover over the Arctic (60°-90°N) for 2000-2014 shows that NCEP data underestimates the mean cloud
cover amount by about 25-30% all year round (Liu and Key, 2016).

For sea-ice, we use the following four different data sets. (i) Sea-ice concentrations (Figure 3) are derived from the
Advanced Microwave Scanning Radiometer for EOS (AMSR-E) with errors less than 10% for ice concentrations above 65%
(Spreen et al., 2008). The spatial grid resolution for ice concentration is 6.25 km. The time series of sea-ice concentration
was obtained by averaging the daily data at all grid nodes over the eastern Beaufort Sea area limited by 220°E to 230°E and
70°N to 72°N. (ii) The long-term mean annual cycles of the first- and multi-year pack ice draft were adapted from Melling et
al. (2005) and Krishfield et al. (2014), respectively (Figure 3). Melling et al. (2005) compiled the mean annual cycle in the
sea-ice draft of the first-year pack ice using data from the Ice Profiling Doppler Sonar (IPS) by ASL Environmental Sciences
deployed over the eastern Beaufort Sea shelf between 1991 and 2003 at about 50 m depth, 156 km southwestward from the
CA13 mooring. Krishfield et al. (2014) computed the mean annual cycle in the multi-year pack ice draft using the IPS data
from the Canada Basin. We used their data from the IPS at mooring BGOS-D (74°N, 220°E) deployed from 2006 to 2012
about 400 km northwestward from the CA13 mooring. (iii) We also used data on sea-ice thickness from the Ice, Cloud, and
land Elevation Satellite (ICESat) obtained through https://rkwok.jpl.nasa.gov/icesat/index.html (Figure 4). Data represent the
gridded 25-km means. Kwok et al. (2007) found a mean uncertainty of the sea ice thickness of about 0.7 m, and the sea-ice
draft estimated from ICESat data relative to that measured at moorings agreed within 0.5 m. We use data from the ICESat
campaigns previously used by Kwok et al. (2009): ON03 (24 September - 18 November 2003), FM04 (17 February – 23





March 2004), ON04 (3 October – 8 November 2004), and FM05 (17 February – 24 March 2005) shown in Figure 4. Finally, we used (iv) satellite synthetic aperture radar (SAR) imagery acquired by Canadian RADARSAT over the mooring location before the sea-ice breakup in 2004 and 2005 (Figure 5).


Zooplankton samples were collected in September 2016 at six stations in the southeastern Beaufort Sea (Figure 1a) from the CCGS *Amundsen*. The positions of the stations are the following: - St. 405: 70.606°N, 236.964°E, st. 435: 71.074°N, 226.356°E, st. 406 - 69.972°N, 237.039°E, st. 437: 71.801°N, 233.501°E, st. 403: 70.100°N, 239.893°E, st. 575: 76.181°N, 234.041°E. To identify the taxa involved in DVM, we compared 3 stations sampled during nighttime, at times as

close as possible to local midnight, with 3 other stations sampled during daytime at times close to midday. A double square net, fitted with two 1 m² aperture nets (mesh size 500 and 750 μm), was obliquely deployed at a maximum sampling depth of about 90 m and at a ship speed of two knots. The duration of the plankton net tow varied between 4.5 and 9.8 min. Flowmeters at the mouth of each net were used to calculate the volume of water filtered during net deployment. Upon recovery, fish larvae were rapidly sorted out of the samples. Then, the sample from the 500-μm mesh net was preserved in a

borax-buffered seawater solution of 4% formaldehyde for taxonomic identification. In the laboratory, each sample was sieved through a 1000-μm sieve to retain the zooplankton to be identified. The sample was transferred into a large transparent dish from which all the macrozooplankton organisms (mainly, jellyfish, large chaetognaths, amphipods and euphausiids) were sorted, counted and identified to species level. Known volume aliquots containing approximately 300 organisms were taken from the remainder of the sample using a Motoda splitting box and the zooplankton were counted and

identified to the lowest taxonomic level possible.

## 3 Methods

We analyzed the acoustic backscatter and velocity time series from ADCP to reveal modifications of the acoustic backscatter diurnal signal primarily dominated by DVM. In general, the particles in the water column producing a significant portion of acoustic backscatter comprise suspended sediments or planktonic organisms (e.g., Petrusevich et al., 2020). Frazil ice

crystals also generate an enhanced acoustic backscatter (e.g., Dmitrenko et al., 2010). However, sound scattering produced by zooplankton is more complex compared to that generated by sediment particles due to DVM (Stanton et al., 1994). Moreover, ADCPs, unlike echo-sounders, are limited in deriving accurate quantitative estimates of zooplankton biomass (Lemon et al., 2001, 2012; Vestheim et al., 2014). This is mainly due to calibration issues (Brierley et al., 1998; Fielding et al., 2004; Lemon et al., 2008; Lorke et al., 2004) and the beam geometry (Vestheim et al., 2014). To account for the beam

geometry, we derived mean volume backscatter strength (MVBS) in dB from the acoustic backscatter echo intensity following the procedure described by Deines (1999).

Under-ice illumination (Figures 6b and 7b) was modelled using the exponential decay radiative transfer model (Grenfell and Maykut, 1977; Perovich, 1996). Transmittance through the sea ice and snow cover to depth $z$ in the ice was calculated using the following equation: $T(z)= i_0e^{-K_tz}$, where $i_0$ is the fraction of the wavelength-integrated incident irradiance

transmitted through the top 0.1 m of the surface layer, and $K_t$ is the total extinction coefficient in the snow or sea ice cover.





The values adopted for the sea ice and snow covers were $i_0$=0.63, $K_t$=1.5, and $i_0$=50.9, $K_t$=0.1, respectively (Grenfell and Maykut, 1977). For computing under-ice illumination in Figures 6b and 7b, we use the mean (2006-2012) annual cycle derived from the IPS data by Krishfield et al. (2014) – Figure 3. The snow thickness on the top of the ice was gradually increasing from zero at freeze-up to a typical 15-cm thick wind-packed snow in late winter (Melling et al., 2005).

Time series of MVBS (Figures 6c-g), vertical velocity component (Figures 7c-g) and surface layer illumination (Figures 6b and 7b) are presented in a form of actograms showing a rhythm of activity. The diurnal signal variation is presented along the vertical axis of the actogram, while the long-term patterns of diurnal behavior can be assessed following the horizontal axis (e. g., Leise et al., 2013; Last et al., 2016; Hobbs et al., 2018). In Figures 6b and 7b we introduced an artificial visual boundary on the illuminance colour scheme at 1 lux (gray to orange), the threshold, which corresponds to illuminance during
the deep twilight.

Following Barber et al. (2015), we used the kinetic energy, $E = (U^2 + V^2) / 2$, derived from the zonal ($U$) and meridional ($V$) components of the current velocity to identify the major energetic events exceeding the two standard deviation threshold of ~ 500 cm$^2$ s$^{-2}$. Using this threshold, Dmitrenko et al. (2018) identified thirteen major energetic events comprised of upwellings and downwellings. They are highlighted in Figures 6-8 with blue and pink shadings, respectively.

## 145 4 Sea-Ice and Oceanographic Settings

### 4.1 Sea-ice

The southern Beaufort Sea is seasonally ice-covered. It is dominated by the first-year pack ice with thickness gradually increasing from zero in September to ~80-90 cm in March-April (Melling et al., 2005) – Figure 3. In the Canada Basin beyond the eastern Beaufort Sea continental slope, ice conditions are partly dominated by the multi-year pack ice with a
mean thickness increasing from about 30 cm in August-September to 210-220 cm in May (Krishfield et al., 2014) – Figure 3. The multi-year Greenland pack ice (>7 m thick) occupies the area to the north of the Canadian Arctic Archipelago and Greenland (e.g., Kwok et al., 2009).

During winter 2005, the on-slope displacement of the multi-year pack ice from the Greenland and Ellesmere Island shelves disrupted the long-term mean seasonal cycle. This is evident from the sea-ice thickness ICESat data showing a west-
southward expansion of the Greenland pack in February - March 2005 (Figure 4d). This is in line with detecting multi-year ice on the RADARSAT satellite imagery acquired over the mooring position (Figure 5). The lighter areas in Figures 5c and 5d indicate the multi-year pack ice expanded over the mooring position before the sea-ice breakup in May 2005. Overall, satellite data show that during winter-spring 2005 sea-ice thickness over the mooring location exceeded that for 2004 by >1 m suggesting implications for the under-ice illuminance values.





## 4.2 Temperature and salinity

The structure of the near-surface and intermediate water over the eastern Beaufort Sea upper continental slope, resolved by ADCP, is comprised of a mixture of river runoff and sea-ice meltwater and seawater of Pacific origin (Figure 2). A surface layer of relatively warm and low-salinity water (~27-29) is freshened by the Mackenzie River runoff and sea-ice melt. Water with the salinity 29 < S < 33 is generally assigned to Pacific Water (PW) – e.g., Dmitrenko et al. (2016). It is transported along the Beaufort Sea continental slope by an Alaskan branch of the PW flow emanating from Bering Strait. The relatively fresh PW layer impacts the halocline structure, producing a double halocline layer with a low stratified Upper Halocline water formed by the insertion of PW that overlies Lower Halocline water originating from the Eurasian Basin. In this study, we associated PW with the broad temperature range between 1.5°C and –1.5°C approximately centered at S ~ 32 residing upper and low halocline layers (Figure 2). Pacific Summer Water (PSW) is broadly classified here as T > –1.2°C and 30 < S < 32 (pink shading in Figure 2). In October 2003, July 2004, and September 2005, PSW occupied the upper intermediate water layer from ~25 to ~60 m depth (Figure 2). This water mass is usually comprised of the Chukchi Summer water transported through Herald Canyon on the western Chukchi shelf (Woodgate et al., 2005) and the Alaskan Coastal water transported by the Alaskan coastal current through Barrow Canyon (Pickart et al., 2005). The underlying Pacific Winter Water (PWW) with the broad temperature minimum below –1.2°C centred at S ~ 33 (blue shading in Figure 2) is generated during freezing and brine rejection in the Bering and Chukchi Seas (Weingartner et al., 1998; Pickart, 2004). During 2003-2005, PWW occupied the lower intermediate water layer in ~60-140 m depth (Figure 2). The warm and saltier Atlantic water with temperatures above 0°C and S > 33.5 underlies PWW at depths > 230 m that significantly exceeds the depth range resolved with ADCP measurements (Figure 2).

## 4.3 Water dynamics

The kinetic energy of currents over the eastern Beaufort Sea continental slope is mainly affected by the along-slope current component (Kulikov et al., 1998; Williams et al., 2006; Dmitrenko et al., 2016, 2018). For CA13, the maximum variability of currents is also consistent with along-slope direction, explaining ~ 70% of the total velocity variability (Dmitrenko et al., 2018). Thus, major energetic events highlighted in Figures 6-8 are primarily associated with along-slope flow dynamics, as also follows from the velocity time series in Figures 8c and 8d. Among thirteen major energetic events in Figures 6-8, four events were clearly attributed to the depth-intensified flow (#3D, 4D, 6D and 10D; pink shading in Figures 6-8) generated by ocean downwelling superimposed on the background bottom-intensified eastward shelfbreak flow. Six events are associated with the surface-intensified or barotropic flow (#1U, 2U, 7U, 8U, 9U, and 12U; blue shading in Figures 6-8). These events were attributed to ocean upwelling (Dmitrenko et al., 2018). While events #5U and 11U are depth-intensified, they are highlighted with blue shading because they are consistent with upwelling-favorable atmospheric forcing that usually drives the surface-intensified events. Vice versa, event #13D is surface-intensified, but it has been highlighted with pink shading because it is consistent with downwelling-favorable atmospheric forcing (Dmitrenko et al., 2018).



## 5 Diurnal Signal of the Mean Volume Backscatter Strength (MVBS) and Vertical Velocity

MVBS and vertical velocity actograms were computed for the depth of 28, 48, 68, 88 and 108 m (Figures 6c-g and 7c-g). These actograms reveal a rhythm of activity with diurnal signal variations seen in the vertical axis of an actogram. The 2-

year long variability of the diurnal pattern is observed along the horizontal axis.

### 5.1 Seasonal patterns

In general, MVBS actograms resemble seasonal variability of the diurnal signal following light conditions (Figures 6b-g and 7b-g). In the subsurface layer (28 m depth), a low MVBS corresponds to a relatively high illuminance during the day, while an elevated MVBS is consistent with a low illuminance during the night (Figures 6b and 6c). In contrast, at 108 m depth,

MVBS shows an opposite pattern with a high MVBS during the light time of the day and a low MVBS in the darkness (Figures 6b and 6g). This variability in MVBS is consistent with DVM.

The main factor controlling surface and under-ice illuminance is solar altitude. Civil polar night (the sun is between 6° and 12° below the horizon for the entire 24 hours) does not occur for CA13 at ~71.3°N. Civil twilight (the sun is down to 6° below the horizon) is observed at the CA13 latitude from 18 November to 21 January (Figures 6b and 7b). For the winter

solstices (22 December), the civil twilight lasts for about 3 h. The polar day (midnight sun) occurs when the sun is above the horizon for the entire 24 hours. At the CA13 latitude, the polar day lasts 82 days from 10 May to 31 July.

In general, the MVBS diurnal signal follows the seasonal variability of the sun illuminance during the entire year except for the period of the polar day when the diurnal pattern becomes significantly disrupted in the sub-surface water layer. Outside of the polar day, the sun illuminance is opposite to MVBS for the sub-surface layer, while at 108 m depth this

relationship becomes positive. During the polar day, in the subsurface layer the MVBS diurnal rhythm vanishes (Figure 6c). In spring 2004, the modification of the MVBS diurnal pattern from the beginning of May corresponds to an increase of the midnight under-ice illuminance to >1 lux (Figures 6b and 6c). This modification lagged behind the sea-ice retreat off the mooring location by about one month (Figures 6a and 6c). In contrast, during spring 2005, significant deviation of the MVBS diurnal rhythm was delayed by about 3 weeks compared to 2004. The deviation of the MVBS diurnal pattern was

recorded once the sea-ice disappeared from the mooring location on 20 May 2005. Note that the satellite-derived data for spring 2005 show the sea-ice thickness over the mooring location exceeding that for 2004 by >1 m (Figures 4 and 5). In spring 2005, the midnight under-ice illuminance >1 lux was lagging that in 2004 by about one week (blue dotted curve in Figure 6b). Note that for winter-spring 2005 the under-ice illuminance, shown in colour in Figure 6b, is overestimated by about a factor of 10 (not shown). Note that the under-ice illuminance has been simulated using the long-term mean seasonal

cycle of sea-ice thickness reported by Melling et al. (2005). The ICESat data set has no capacity for generating the sea-ice thickness seasonal cycle for 2003-2005.

For the PW layer, the behavior of the MVBS diurnal signal during the polar day is different from the sub-surface water layer. From 1 April to 10 July 2004, the MVBS diurnal signal was enhanced at the 48-108 m depth layer (Figures 6d-



6g). In contrast to the preceding and subsequent periods, no seasonal tendency in the duration of high/low MVBS was
observed at this time. This is in line with illuminance, showing almost no seasonal modulation during the midnight sun
(Figure 6b). For the polar day period in 2005, however, enhancement of the MVBS diurnal signal seemed to be masked by
the short-term high MVBS events likely generated by intrusions of turbid water. These events were found to be most
pronounced through the PSW layer where intrusions of turbid and relatively warmer water were observed during mooring
recovery (Figures 2c, 6d, and 6e).

Following the midnight sun, the MVBS diurnal signal returned once the mooring position becomes 100% ice-
covered since 7 November 2003 and 25 October 2004 (Figures 6a and 6c-6g). This is evident from enhancing the MVBS
difference between the light (> 1 lux) and dark (< 1 lux) time for 28-48 m depth (Figure 6c-6d). The noticeable feature of the
MVBS diurnal signal during civil twilight and the subsequent period until the end of April is a significant MVBS difference
between 2003-2004 and 2004-2005 observed during the dark time through the entire water column resolved with ADCP
observations (Figures 6c-6f). Another noticeable feature of MVBS during this period is numerous disruptions of the diurnal
signal discussed below in sections 3.2.2 and 3.2.3.

Behind seasonality of diurnal signal in the MVBS time series, the seasonal cycling of the MVBS vertical
distribution has been revealed (Figures 8a and 8b). For midnight, during low-light conditions from October to February (civil
twilight length exceeds daylight length), MVBS tends to increase with depths from 28 to 108 m depth (Figure 8a). In March-
June, the midnight MVBS shows an opposite tendency (Figure 8a). The MVBS midnight long-term mean, however, shows
almost no difference from 28 to 108 m depth with a long-term mean of -0.6 dB. The seasonal cycle of the MVBS vertical
distribution for the astronomic noon is different. From about winter to summer solstices, MVBS at 128 m depth exceeds that
for 28 m depth by about 8 dB (Figure 8b). In contrast, during the ice-free period in June-August, the MVBS difference from
28 to 108 m depth tends to decrease down to about zero in late summer. The long-term mean for the astronomic noon (-5.3
dB, Figure 8b) shows a general tendency of MVBS to increase with depth.

The vertical velocity actograms also show a diurnal pattern around astronomic midnight (Figures 7c-7g) that is
consistent with the MVBS diurnal rhythm in Figures 5d-5h. Net upward movement is regularly observed before the
astronomic midnight once the under-ice dark-time illuminance was <1 lux (Figures 7b-7g). Moreover, the most intense
upward flow was recorded during 1-3 h after the illuminance dropped below the 1 lux threshold. In contrast, a downward net
flow was recorded following the tendency of under-ice illuminance to increase from midnight to noon once illuminance
exceeds the 1 lux threshold (Figures 7b-7g). At the end of April 2004, once under-ice illuminance exceeded the 1 lux
threshold for 24 h a day approaching the midnight sun, the vertical velocity diurnal signal completely vanished. During May-
June 2004, however, a weaker net upward and downward diurnal movement of about ±0.5 cm s-1 was recorded at 68 and 88
m depth from noon to midnight (light blue to green shading in Figures 7e and 7f), and from midnight to noon (light green to
yellow shading in Figures 7e and 7f), respectively. This is consistent with the MVBS diurnal rhythm revealed through the
PW layer during summer 2004 (Figures 6e-6g). Following the under-ice illuminance, well-pronounced velocity diurnal
signal again appeared since mid-August 2004 when the midnight under-ice illuminance decreases to the 1 lux threshold



gradually returning to civil twilight. In spring 2005, the vertical velocity diurnal signal was relatively well pronounced until the midnight under-ice illuminance is below the 1 lux threshold (Figures 7b-7g). As of MVBS, complete cessation of a diurnal signal in vertical velocity in spring 2005 was observed at 68-88 m depth only when sea-ice started to retreat in mid-May (Figure 7a). In this case, complete cessation of diurnal signal lagged the 1 lux threshold by about 10 days (Figures 7b, 7e, and 7f). During the midnight sun 2005, the velocity diurnal rhythm is unrecognizable.

Finally, the velocity diurnal signal varies with depth. The upward and downward flow attributed to diurnal cycling is higher and less noisy at 68-88 m depth compared to the overlaying sub-surface layer at 28-48 m depth and to a lesser extent to the underlying water at 108 m depth (Figures 7c-7g).

## 5.2 Moon cycle

During the period of civil twilight when the sun is more than 6° below the horizon, the moonlight was the main source of illumination over the eastern Beaufort Sea continental slope (Figure 6b). The full moon succeeds with a mean period of 29.53059 days called a synodic or lunar month. During winter, the full moon generates under-ice illuminance up to about 0.1 lux below the sea ice layer with a thickness of around 0.5 m (Figures 3 and 6b). While the cloud cover attenuates moon illumination, it was not considered for modelling under-ice illuminance.

The MVBS diurnal signal is impacted by the moonlight, also attenuated by the cloud cover. Once the full moon (±6 days) occurred during the period of civil twilight, the cloud cover shows three low-cloud events with cloud cover ≤ 30% (#3, 4 in 2003-2004 and #3 in 2004-2005 in Figures 6a and 6b). During these events, the MVBS diurnal signal was significantly disrupted in the sub-surface layer, and a low MVBS was observed during the entire 24 h (Figures 6bc and 6c). For the full moon event #4 in 2004, during the astronomic midnight, a low MVBS at 28 m depth was associated with an elevated MVBS at 108 m depth, as evident from decreasing the MVBS difference from 28 to 108 m depth in Figure 8a. Overall, among 14 full moon events occurred in October-March 2003-2004 and 2004-2005 once the midnight under-ice illuminance was <1 lux, 10 events demonstrated similar, but less intense anomalies of the MVBS difference (events #1-3, 4,5, and 7 in 2003-2004 and #1, 2, 5 and 7 in 2004-2005, Figure 8a). During the noon, however, this pattern is not obvious (Figure 8b).

Full moon event #1 in September-October 2004 gives an example of the moonlight impact on the MVBS diurnal signal (Figure 6). While the cloud cover during this event was relatively high (~50%, Figure 6a), the dark-time MVBS dropped by ~2 dB at 28 m depth, but elevated by ~4 dB at 68 and 88 m depth suggesting the downward displacement of the acoustic backscatters (Figures 6c and 6f-6g, respectively). At noon, however, MVBS elevated by ~4 dB at 28 m depth (Figure 6c). Note that during this time the under-ice illuminance reduced as the mooring became ice-covered (Figure 6a). It is also important to point out that this full moon event partly overlays with upwelling #7U described below.

## 5.3 Short-term oceanographic events

The regular diurnal pattern of MVBS was disrupted during the short-term events lasting from several days to several weeks (Figures 6d-6h). These events also interplay with disruptions generated by moon cycling. We use actograms of vertical





velocity to differentiate disruptions imposed by moonlight from those of dynamic origin (Figures 7c-7g). In general, the diurnal pattern remains well recognizable during the full moon events (Figures 7b-7g). In contrast, almost all significant or even complete short-term disruptions of the vertical velocity diurnal rhythm are related to upwelling or downwelling (Figure 7).

Dmitrenko et al. (2018) identified upwelling or downwelling events at CA13 using ADCP velocity data, the NCEP-derived wind and sea-level atmospheric data, sea surface height records at Tuktoyaktuk (Figure 1) and numerical simulations. All these events are highlighted in Figures 5-7 with blue and red shadings for upwelling and downwelling, respectively. Vertical displacement of water parcel attributed to upwelling or downwelling generates a noise of vertical velocity deviating its diurnal pattern. These deviations, however, can also be attributed to inclinations of the ADCP transducer due to high-velocity currents. In the following, we are only interested in the vertical velocity estimates, which are sensitive to the MVBS diurnal cycling.

### 5.3.1 Upwelling Events

Upwelling events disrupt the MVBS diurnal signal in a similar way as the moonlight does. For upwelling #1U, MVBS at 108 m depth was elevated throughout the full 24 h period (Figure 6g). During the dark time (illuminance < 1 lux) at 28 m depth, MVBS reduced to the end of the event when the surface-intensified flow at 28 m depth shows maximum velocities exceeding 30 cm s$^{-1}$ (Figure 8c). Moreover, upwelling #1U resulted in ~0.7°C temperature increase at 119 m depth (Figure 8f). Upwelling #2U, occurred right before the winter solstice, shows significant MVBS reduction at 28-48 m depth gradually vanishing to 108 m depth. Upwelling #5U occurred at the end of the ice-free season and shortly after the end of the midnight sun 2004. Therefore, the MVBS diurnal signal was relatively weak and noisy, especially at 28 m depth. However, MVBS increase at 88-108 m depth is likely attributed to upwelling. Upwelling #7U interplayed with full moon event #1 in September-October 2004. It seems that the first portion of this event until 3 October 2004 was dominated by the moonlight. Afterward, once the horizontal velocity at 28 m depth exceeded ~30 cm s$^{-1}$ (Figure 8c), a slight reduction of MVBS is observed in 28-48 m depth during the dark time. In contrast to the preceding upwelling events, no elevated MVBS values were recorded in the overlying water layer. Upwelling #8U completely coincided with full moon event #2 in October-November 2004. As with the majority of the full moon and upwelling events, it shows the downward redistribution of the acoustic backscatters from 28-48 m to the deeper water layer (Figures 6c-6g, 8a, and 8b). A similar overlap between full moon and upwelling was observed during upwelling #9U. Significant MVBS reduction within 28-48 m was accompanied by elevated MVBS in 88-108 m depth during the latter part of this upwelling from 25 November to 5 December 2004. Overall, upwelling events #7-9 resulted in a gradual increase of temperature at 119 m depth from -1.55°C to -0.65°C (Figure 8f). Upwelling #11U shows an elevated MVBS during the light time at 88-108 m (Figures 6f, 6g, and 8b). However, no significant modifications of the MVBS diurnal signal were observed in the overlying water. The last upwelling #12U in May-June 2005 occurred during the midnight sun when the MVBS diurnal signal mostly vanished, and MVBS is noisy. We speculate that this noise is due to the enhanced concentration of suspended particles in the water column (Figure 2c).



Overall, among 8 upwelling events observed in 2003-2005, 6 events (#1, 2, 5, 7, 8, 9U) clearly show the MVBS reduction in the subsurface water layer at 28 m depth (Figure 6c). For upwellings #1, 5, 7, and 8U the midnight MVBS

difference from 28 to 108 m depth tends to decrease, which is consistent with a downward redistribution of acoustic scatters (Figure 8a). This effect is similar to the MVBS response to the full moon events as described in section 5.2. It seems that the overlay between the upwelling and full moon can dominate the MVBS response to upwellings #7-9U (Figures 6b-6g). During the polar day, the MVBS diurnal signal is weak or completely disrupted, and its response to upwelling is barely traceable (upwelling #12, Figure 6b-6g).

**5.3.2 Downwelling Events**

Downwelling events disrupt the MVBS diurnal signal in the opposite way compared to upwellings and moonlight, moving MVBS upwards. Downwelling also interferes with MVBS modifications imposed by sea-ice and the MVBS diurnal signal deviations generated by moonlight. Wind, forcing downwelling events, also impacts the sea-ice cover through the on-shelf displacement of the pack ice as evident for downwelling events #4, 6 and 13D (Figure 6a). Deviations of the MVBS diurnal

signal due to moon cycling interferes with those caused by downwelling event #3 complicating our analysis.

Downwelling #3D occurred at the end of 2003-2004 during civil twilight and strongly interfered with full moon event #4 (Figure 6). It seems that downwelling #3D is entirely dominated by the moon, disrupting the MVBS diurnal signal as described in section 5.2. Downwelling #4D was recorded at the end of the polar day 2004 when the MVBS diurnal signal is terminated at 28-88 m depth (Figures 6c-6f). Wind, forcing downwelling #4D, displaced pack ice on-shelf, and the CA13

position was reoccupied by sea-ice for about 10 days with implication for under-ice illuminance. Figures 6c and 6d show that sea-ice and downwelling did not impact MVBS at 28-48 m depth. In contrast, the midnight sun diurnal signal at 68-88 m depth was disrupted due to elevating MVBS at 68-88 m depth during the dark-time (from 22 to 4 h, Figures 6e and 6f). At the same time, the midnight sun diurnal signal at 108 m depth remained undisturbed (Figure 6g). Downwelling #6D provides the most comprehensive example of how the MVBS diurnal signal is disrupted by downwelling. In contrast to the full moon

and upwelling events, MVBS at 28-48 m depth was enhanced 24 h a day (Figures 6c and 6d) suggesting the upward redistribution of the acoustic backscatters from the underlying water layer (Figures 8a and 8b). Note that during this event water dynamics was dominated by along-slope depth-intensified flow increasing from ~5 cm s$^{-1}$ at 28 m depth (Figure 8c) to >30 cm s$^{-1}$ at 108 m depth (Figure 8d). Downwelling #10D was recorded at the end of civil twilight 2004-2005. It appears that the beginning of this event is impacted by the full moon (#4, 2004-2005) with the reduction of MVBS in the sub-surface

layer at 28 m depth. However, by the end of downwelling #10D, once the bottom-intensified flow exceeded 100 cm s$^{-1}$ at 128 m (Figure 8d), MVBS at 28-48 m depth tended to increase suggesting the upward redistribution of the acoustic backscatters, similar to downwelling #6D. Downwelling #13D occurred in mid-August 2005 following the midnight sun. During this time, the MVBS diurnal signal at 28 depth was not traceable. At 48-88 m depth, the midnight sun diurnal signal was likely masked due to the enhanced concentration of suspended particles in the water column (Figure 2c).





Overall, among 5 downwelling events recorded in 2003-2005, event #6D and partly #10D show disruption of the MVBS diurnal signal in the sub-surface water layer with MVBS elevated at 28-48 m depth in response to downwelling. Downwelling events #4 and 13D occurred during and shortly after the midnight sun, respectively, when the sub-surface MVBS diurnal signal vanishes. Downwelling event #3D was dominated by the moonlight.

**5.3.3 Eddies**

Eddies are ubiquitous over the Arctic Ocean continental slope (e.g., Dmitrenko et al., 2008; Pnyushkov et al., 2018), and particularly over the Beaufort Sea continental slope (e.g., Spall et al., 2008; O'Brien et al., 2011). The eddy carrying entrained suspended particles was identified by Dmitrenko et al. (2018) based on the ADCP velocity and acoustic backscatter time series in February–March 2004 (Figures 6 and 9). One more eddy passed mooring position in December/January 2003/2004 right before downwelling #3D. In Figures 6-8 both eddies are highlighted with yellow

shading.

        The eddy in February–March 2004 provides an example of how the velocity field attributed to eddy passing disrupts the MVBS diurnal signal (Figure 9). The greatest tangential speed, exceeding 22 cm s$^{-1}$, marks the eddy core near 95 m depth (Figures 9a and 9b). Below the core at 119 m depth, a positive temperature anomaly of 0.25°C attributed to the eddy passing was recorded on 26 February 2004 (Figure 8f). The velocity signature of the eddy is hardly discernible shallower

than about 50 m, where the temperature anomaly does not exceed ~0.1°C (Figures 8e, 9a, and 9b). During the dark time at 108 m depth (below the depth of the greatest tangential speed), an enhanced MVBS was observed between two maximal of the eddy tangential speed from 27 February to 2 March 2004 (Figure 9d). In contrast, during the daylight, a negative MVBS anomaly was recorded (Figures 6g and 9d). This completely inverted the MVBS diurnal signal observed at 108 m depth during the eddy passing. At 28-48 m depth, however, MVBS was not modified (Figure 9c). In contrast to water layers above

and below the eddy core, from 26 February to 1 March the MVBS diurnal signal at 68-88 m depth was disrupted by the backscatter maximum recorded for 24 h a day (Figures 6e and 6f). It appears that this MVBS anomaly is attributed to the eddy-entrained suspended particles commonly recorded in this area (O'Brien et al., 2011).

        The eddy in December/January 2002/2003 generated much less MVBS disturbance compared to the one in February–March 2004 (Figures 6c-6g). The core of the eddy was likely deeper than the ADCP transducer. A positive

temperature anomaly at 119 m depth was 0.5°C (Figure 8f). A positive MVBS anomaly was recorded only at 108 m depth during 24 h a day (Figure 6g) likely indicating the eddy-entrained suspended particles. MVBS in the overlaying water layer was not significantly modified.

        In summary, the eddy in February–March 2004 inverted the MVBS diurnal signal in the water layer below the eddy core defined by the greatest tangential speed of the horizontal flow. The eddy in December/January 2002/2003 generated no

significant MVBS modifications as the eddy core was likely located below the ADCP.





## 6 Zooplankton

The assemblage of organisms making the large mesozooplankton size class (>1 mm) was more abundant at night (34-65 ind. $m^{-3}$) than in the day (4-12 ind $m^{-3}$) within the top 90 m of the water column. This difference in zooplankton's total abundance between day and night was mainly due to the higher abundance of the large copepods *Calanus glacialis* and *Metridia longa*

at night (Figure 10a).

The abundance of the macrozooplankton size class was much lower than that of the large mesozooplankton, representing between 0.3% and 8% of the total zooplankton assemblage. However, the difference of total macrozooplankton abundance between night and day was not as clear as for the large mesozooplankton size class. No species stood out as being more abundant at night than at day (Figure 10b).

## 7 Discussion

The two-year-long ADCP time series of MVBS and vertical velocity over the upper eastern Beaufort Sea continental slope are consistent with DVM of zooplankton. MVBS diurnal signal is generated by a diurnal movement of zooplankton toward the surface at dusk, and descent back the next morning before dawn. DVM demonstrates predator-avoidance behavior (Hays, 2003). Zooplankton keeps away from a relatively well-illuminated surface water layer during the light time reducing the

light-dependent mortality risk.

In general, DVM at CA13 is controlled by light conditions (Figures 6 and 7). As for the other areas of the ocean, DVM is triggered by local solar variations, and the timing of migration is sensitive to changes in seasonal day length (e.g., van Haren and Compton, 2013). Our results show that DVM responds to (i) seasonality of the sunlight, (ii) moonlight, and (iii) seasonality of sea-ice cover that attenuates light transmission to the water column. Moreover, (iv) DVM can be modified

or completely disrupted during highly energetic current events generated by upwelling, downwelling or eddy passing. Our results also suggest that the interplay between all these factors impacts DVM at CA13. Furthermore, MVBS is not entirely controlled by DVM. The suspended particles in the water column enhance acoustic scattering, masking DVM during the midnight sun (Figures 2a, and 6b, 6d, and 6e), and also attenuating light intensity in the water column. Below we discuss all these factors and their impact on DVM in more detail.

### 7.1 Zooplankton species generating DVM

The acoustic data from the single-frequency ADCP do not provide any information on the identity of organisms responsible for the observed DVM patterns, and proper studies on DVM have not been carried out in the Beaufort Sea prior to the present work. Differences in the zooplankton composition and abundances at night and in the day, revealed by plankton net sampling in the surface layer of sampling stations around the location of the mooring, indicate that the large copepods

*Calanus glacialis* and *Metridia longa* were performing DVM in September. Their density in the surface layer was higher at night than at day in September 2016. The advanced developmental stages (copepodites CIV-adults) of these species have





also been found to perform DVM in the other parts of the Arctic, such as Barrow Strait in the central Canadian Arctic Archipelago in May and June (Fortier et al., 2001), and around the Svalbard Archipelago in the European Arctic in September (Daase et al., 2008). Among the large copepods in the Svalbard Kongsfjord, only *M. longa* performed DVM in

February at the termination of the polar night (Grenvald et al., 2016). *Calanus glacialis* and *M. longa* CIV-adults are >2.2 mm in length, providing them with good backscatter potential at the 300 kHz frequency of the ADCP (Berge et al., 2014). Thus, they were likely detected by the ADCP and contributed to the diel signals recorded.

Grenvald et al. (2016) also found that the euphausiids *Thysanoessa* spp. had the highest backscatter potential in February in Kongsfjord, partly due to their large macrozooplankton size, and were most likely responsible for the DVM

patterns observed at that time. Although, *Thysanoessa* spp. were found in our zooplankton samples, their abundance, or that of any other euphausiids, was very low in September 2016 at the six sampling stations considered in our study. In fact, the lack of convincing day-night difference in the abundances of any of the taxa belonging to the macrozooplankton size class is not necessarily indicative of an absence of DVM behaviour on their part. Our sampling method, mostly designed to capture small fish larvae, may just have not been efficient enough for quantitative sampling of fast-swimming macrozooplankton

crustaceans such as pelagic amphipods and euphausiids, known to perform DVM in other regions. Furthermore, comparison of day and night plankton net samples from different locations may introduce variability, which is difficult to characterize due to a limited number of stations used in our study. A more thorough approach to properly identify the zooplankton species engaged in DVM should rely on much larger plankton nets, and day-night sampling events at the same locations over several stations to account for spatial difference and patchiness. This would represent a significant logistical challenge in the

remote Arctic marine systems. Moreover, the deficiency of our analysis clearly defines a necessity for expanding mooring observations using underwater electronic holographic cameras such as those described by Sun et al. (2007).

**7.2 DVM seasonal cycle, sea-ice cover, and suspended particles**

It appears that DVM is triggered once the estimated near-surface illuminance falls below the 1-lux threshold. This suggests that the diurnal movement of zooplankton toward the surface at dusk starts once the near-surface illuminance decreased to

~1 lux, and descends back the next morning before dawn as soon as the near-surface illuminance exceeds 1 lux. DVM follows changes in seasonal day length, and it stops at the sub-surface layer as soon as near-surface illuminance retains above ~1 lux for 24 h a day (Figures 6b-6g). At the CA13 latitude (71°21.356'N), the estimated value of near-surface or under-ice illuminance exceeds the 1 lux threshold for about 12 and 7 days before the midnight sun in 2004 and 2005, respectively (Figure 6b). During fall 2004, the sub-surface DVM returned about 20 days after the polar day season, once the

midnight near-surface illuminance dropped below ~1 lux threshold around 22 August (Figures 6b-6g). The inter-annual variability in estimated under-ice illuminance is entirely attributed to the sea-ice thickness. During the ice season, the mean cloud cover (~40%) showed insignificant interannual variability (Figure 6a); thus, the cloud cover was not taken into account.





Our results reveal that sea-ice cover modifies the DVM seasonal cycle by attenuating under-ice illuminance. Before
the beginning of the polar day 2004, CA13 was covered with the first-year pack ice of about 1.6 m thick (Figures 4b, 5a, 5b).
In contrast, during the same time in 2005, and likely during the entire winter 2004-2005, the eastern Beaufort Sea continental
slope was occupied by the multi-year pack ice of about 2.6 m thick (Figures 4c, 4d, 5c, and 5d). We suggest that this
increased sea-ice thickness extended the DVM seasonal cycle toward the polar day of 2005. In spring 2005, the 1-lux
threshold estimated for 2.6 m thick ice lags that for 2004 by about one week (Figure 6b). Following ice-diminished
illuminance in April-May 2005, DVM at 28 m depth was clearly recorded until 7 May 2005. Moreover, DVM maintained
integrity at 68-108 m depth until the open-water season started in mid-May 2005 (Figures 6c and 6e-6g, respectively). In
contrast, during spring 2004, DVM vanished about 10 and 30 days ahead of the polar day and open-water season,
respectively (Figures 6). We suggest that this inter-annual DVM variability is consistent with under-ice illuminance. Its
estimated value for May 2004 (≥10 lux, Figure 6b) exceeds that for May 2005 by a factor of 10 (not shown).

The MVBS actograms show asymmetry of the DVM seasonal cycle to the summer solstice (Figures 6b-6g). In
summer 2004, the DVM seasonal cycle terminated about 54 days before the summer solstice but resumed, lagging summer
solstice by 61 days. This asymmetry, being consistent with the estimated 1-lux threshold, is likely attributed to the seasonal
sea-ice cover. During spring, the polar day begins when the eastern Beaufort Sea continental slope is still ice-covered
(Figures 6a and 6b), which governs attenuation of light below the ice. In contrast, after the polar day is ended, the eastern
Beaufort Sea continental slope remains ice-free or partly ice-covered until the end of October allowing sunlight to illuminate
the near-surface water layer.

In the subsequent winters, the DVM backscatter intensity shows significant interannual variability. The dark-time
MVBS during winter 2003-2004 exceeds that for winter 2004-2005 by ~ 3-5 dB (Figures 6c-6g). We atribute this inter-
annual variability to attenuation of light by a thicker ice cover in winter 2004-2005, as follows from our preceding
discussion. An assumption that the eastern Beaufort Sea continental slope was occupied by Greenland pack ice during the
entire winter 2004-2005 leads to the reduced estimate of under-ice illuminance by a factor of 10 (not shown).

In general, our results on the sea-ice impact on DVM show that DVM is well synchronized with the light/dark cycle
modified by the sea-ice cover shading. It appears that thicker ice observed during winter 2004-2005 reduced the backscatter
values (Figures 6c-6g), which likely demonstrates a light-mediated response of the zooplankton involved in DVM. This is in
line with Berge et al. (2009) reporting a stronger polar night DVM in the ice-free Svalbard fjord compared to the ice-covered
fjord. Vestheim et al. (2014) reported on shallowing DVM in Oslofjord in response to the freeze-up and subsequent
snowfalls. They attributed this shallowing to a relative reduction of light intensities, which is similar to that observed over
the eastern Beaufort Sea continental slope during spring 2005. La et al. (2015) suggested that sea-ice diminishes DVM
signals by blocking the detectable light intensity for DVM with depth during the Antarctic winter. At the same time, our
results contrast with the observations of Wallace et al. (2010). They found no difference in time of the DVM onset and
cessation between the seasonally ice-covered and ice-free Svalbard fjords, insisting on the role of the relative change in
irradiance for triggering DVM. This discrepancy highlights an important difference between sea-ice in the Svalbard fjords





and the eastern Beaufort Sea continental slope. Rijpfjorden in Svalbard is seasonally ice-covered with land-fast ice of ~0.8 m thick (Wallace et al., 2010). In contrast, in spring 2015 the eastern Beaufort Sea continental slope was occupied by 2.6 m thick multi-year Greenland pack ice (Figure 4d) favoring the synchronized DVM to extend toward the midnight sun.

Our data show that, during the midnight sun 2004, DVM ceased only at 28 m depth. In the underlying PW layer at 48-108 m depth, DVM continued until the beginning of July 2004 (Figures 6c-6e). However, DVM in the PW layer did not occur in phase with the 24-h light cycle. It seems that zooplankton was conducting regular synchronized DVM, but it was still avoiding relatively well-illuminated sub-surface water. This is in line with a predator-avoidance behavior during transitional seasons, but without seasonal modulation, because the sun is above the horizon 24 h a day. In fact, zooplankton limits DVM to the PSW water layer with relatively high chlorophyll fluorescence values during late summer (Figure 2). This can indicate high concentrations of phytoplankton (e.g., La et al., 2018), which zooplankton feeds on. The availability of phytoplankton can be an important factor triggering seasonal variability in DVM (e.g., La et al., 2015).

Usually, synchronized DVM stops during the midnight sun, consistent with a predator-avoidance behaviour of zooplankton conducting DVM (e.g., Blachowiak-Samolyk et al., 2006; Cottier et al., 2006; Wang et al., 2015; Darnis et al., 2017). However, Fortier et al. (2001) reported a clear midnight sun DVM in copepods under the spring ice of Barrow Strait at the centre of the Canadian Arctic Archipelago. They argued that absolute light intensity below sea ice decreases to the thresholds at which the feeding activity of fish slows down. Moreover, DVM below 2-m thick ice in the Canada Basin during the midnight sun was recently reported by La et al. (2018). Following Fortier et al. (2001), we speculate that the absolute light intensity through the PW layer at CA13 was below the threshold of predators' perception allowing DVM during the midnight sun 2004. However, the midnight sun DVM was not obvious in 2005.

We suggest that the midnight sun DVM in 2005 was likely masked by the enhanced concentration of suspended particles through the PSW layer. Suspended particles return the ADCP acoustic signal, producing enhanced MVBS 24 h a day. For example, Petrusevich et al. (2020) reported on enhanced MVBS in Hudson Bay recorded by 300 kHz RDI Workhorse ADCP. They attributed this signal to the suspended particles released to the water column during ice melt. In contrast to the vertically synchronized DVM, suspended particles generated noise that masked DVM during the midnight sun 2005 as evident from Figures 6d-6f. On the CTD profile taken in September 2005, high particulate beam attenuation layers around 25 and 50 m depth match temperature maxima up to 1.3°C (Figure 2c). Moored CTD at 49 m depth shows several maxima up to 0.5°C following summer solstice 2005 (Figure 8e). Two temperature maxima in the beginning and mid-June 2005 (Figure 8e) match MVBS maxima at 68-88 m depth (Figure 6f and 6g). This suggests that MVBS maxima in actograms are generated by lateral advection of warm and turbid water layers. The formation of this water is likely attributed to wind-forced vertical mixing over the Beaufort Sea shelf involving surface riverine water heated by solar radiation and enriched with suspended particles. Alternatively, suspended particles can be attributed to resuspension of bottom sediments over the Beaufort Sea shelf. In any case, regardless of the source of suspended particles, their enhanced concentration in the water column during summer 2005 resulted in increased light attenuation (e.g., Hanelt et al., 2001), which potentially disrupted DVM during the midnight sun 2005.



### 7.3 DVM modifications by moonlight

Our MVBS actograms in Figures 6c-6g indicate modifications of DVM during a few days near the time of the full moon. These modifications are consistent with a lack of the upward moving zooplankton during the dark-time. They were observed from October to March including the civil twilight (Figures 6b-6g). The most pronounced moonlight modifications were observed during low cloud cover periods attenuating the moonlight (Figure 6).

The results on the moon's modifications of DVM are consistent with those previously reported for the Arctic and sub-Arctic regions. Moonlight plays a central role in structuring predator-prey interactions in the Arctic during the polar night below the ice (Last et al., 2016). It has been shown that during the polar night the moon's influence on DVM in the Arctic results in the zooplankton downward migration to deeper water for a few days near the time of the full moon (Webster et al., 2015; Last et al., 2016). This is consistent with a concept that the moon phase cycle in the zooplankton migration is a global phenomenon in the ocean as suggested by Gliwicz (1986). As to DVM, the reason for the moon's modification was hypothesized to be a predator-avoidance behaviour against predators capable of utilizing the lunar illumination. Note, however, that during civil twilight 2005 below ~2.6 m thick pack ice, zooplankton responded to the estimated lunar illumination of 0.001 lux (not shown), which is far below the threshold of human and predators' perception. The moon's modification of DVM during the 2005 civil twilight suggests that zooplankton shows extraordinary sensitivity to illuminance (Båtnes et al., 2013; Cohen et al., 2015; Last et al., 2016; Petrusevich et al., 2016).

### 7.4 DVM disruptions related to water dynamics

Our results revealed that water dynamics temporally impact DVM by disrupting the diurnal rhythm. Upwelling affects DVM the same way as moonlight forcing zooplankton to avoid the sub-surface water layer during the dark time of the day. In contrast, downwelling seems to force zooplankton to stay in the upper intermediate water layer (consisted of PSW) for 24 h a day. During downwelling, zooplankton likely avoids the lower intermediate layer comprised by PWW. The eddy disrupts DVM in the water layer below the eddy core inverting the MVBS diurnal signal. It seems that zooplankton prevents crossing the water layer occupied by the eddy core. The general impression is that zooplankton likely avoids enhanced water dynamics.

The characteristic feature of high-energetic events recorded at CA13 is the depth-dependent behavior of the horizontal flow. For upwelling and downwelling over the eastern Beaufort Sea continental slope, this feature is generated by the superposition of the background and wind-forced flow (Dmitrenko et al., 2018). The wind-driven barotropic flow generated by upwelling and downwelling wind forcing is superimposed on the background bottom-intensified shelfbreak current depicted by a dashed line in Figure 10c (Dmitrenko et al., 2018). For the downwelling storms, this effect amplifies the depth-intensified background circulation with enhanced PWW transport towards the Canadian Arctic Archipelago (Figure 11b and 11c, right). For the upwelling storms, the shelfbreak current is reversed, which results in surface-intensified



flow moving in the opposite direction (Dmitrenko et al., 2018 and Figures 11a and 11c, left). The baroclinic eddies over the Beaufort Sea continental slope are likely explained by the shelfbreak current baroclinic instability (Spall et al., 2008).

It appears that upwelling, downwelling, and eddies disrupt DVM by generating a water layer with an enhanced gradient of horizontal velocity. We suggest that zooplankton avoids crossing this interface during diurnal migration, disrupting DVM. For crossing the high-gradient velocity layers, zooplankton has to spend additional energy. However, zooplankton is known for demonstrating a strategy of minimizing energy use while crossing water layers with enhanced water dynamics (Eiane et al., 1998; Basedow et al., 2004; Marcus and Scheef, 2009; Petrusevich et al., 2016, 2020). For

example, Petrusevich et al. (2016) reported on the DVM deviation in an ice-covered Northeast Greenland fjord in response to the estuarine-like circulation generated by a polynya opening over the fjord mouth (Dmitrenko et al., 2015). Overall, we suggest that in addition to the predator and starvation avoidance, the zooplankton beware to cross the high-gradient velocity layers remaining behind or below them, hence disrupting DVM.

It is suggested that upwelling and downwelling disrupt DVM. Zooplankton is transported offshore during upwelling

and shoreward during downwelling (for review see Queiroga et al., 2007). For upwelling, the wind-driven Ekman offshore transport leads to offshore dispersal and wastage from coastal habitats. This is consistent with MVBS reduction recorded in the sub-surface layer during upwelling events (Figure 6c). In fact, zooplankton can adjust migration strategy to avoid off-shore transport, reversing DVM (Poulin et al., 2002a,b). Moreover, zooplankton can avoid sweeping off-shore by upwelling and onshore by downwelling, maintaining preferred depth in the face of converging and downwelling flow (Shanks and

Brink, 2005). DVM can be also impacted by the property of upwelled water. Huiwu et al. (2015) reported that DVM deviation is caused by aggregation of zooplankton in the upper 10-m layer in response to upwelling over the Chukchi Sea shelf northwest of the Alaskan coast. They explained DVM deviation by the nutrient-rich upwelled water, which favors an enhanced light attenuation by heavy phytoplankton. This, in turn, allows zooplankton to spend most of its time at the near-surface water layer.

We speculate that DVM disruptions attributed to upwelling and downwelling are primarily dominated by along-slope transport rather than the cross-slope transport. In addition to enhancing the cross-slope transport, upwelling and downwelling over the Beaufort Sea continental slope strongly modify along-slope transport through generating depth-dependent currents over the continental slope (Figure 11; Dmitrenko et al., 2016, 2018). We suggest that zooplankton avoids crossing horizontal velocity interface, generated by the superposition of wind-driven circulation and along-slope jet. This

strategy is evident from the DVM disruption caused by the baroclinic eddy in February-March 2004. Below the depth of the maximum tangential speed (~90 m), DVM was found to be reversed (Figures 6g and 9). This is consistent with reversing DVM to avoid the upwelling induced off-shore Ekman transport in the Peru-Chile upwelling system (Poulin et al. 2002a,b). The reversed DVM in response to eddy passing clearly shows that zooplankton is capable to adjust its strategy of diurnal migration to avoid enhanced water dynamics.

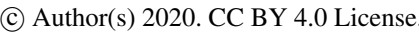



## 8 Conclusions

Based on the 2-year long time series from the mooring deployed over the upper eastern Beaufort Sea continental slope from October 2003 to September 2005, we conclude that the acoustic backscatter is dominated by DVM. DVM is controlled by the following different external forcings that also interplay.

(i) *Illuminance*. It is, in turn, controlled by the solar and moonlight cycling and sea-ice cover. The solar cycle controls DVM and its seasonal variability. In addition, sea-ice modifies seasonal patterns of DVM through light attenuation. A thicker multi-year Greenland pack ice present in winter 2004-2005 reduced the number of acoustic backscatters in the water column compared to that of winter 2003-2004 when the first-year pack ice dominated. Meanwhile, during spring 2005, the multi-year Greenland pack ice favored DVM prolongation toward the midnight sun due to the sea-ice shading the under-ice water layer. During civil twilight, the moon cycle generally modifies DVM, but this modification also depends on the sea-ice thickness and cloud cover. The strongest deviation was observed during mid-fall to early winter when sea ice is absent or relatively thin, and the NCEP-derived cloud cover is <30%. These deviations are associated with significant night-time reduction of acoustic backscatters in the sub-surface layer. Overall, the full moon stimulates zooplankton to avoid the sub-surface layer.

(ii) *Water dynamics*. Upwelling and downwelling disrupt DVM. We found that this disruption is dominated by the along-slope water dynamics rather than the cross-slope Ekman transport. The surface-intensified along-slope flow generated by upwelling drives zooplankton to the lower intermediate depths hosting PWW to avoid the sub-surface layer. Zooplankton similarly respond to upwelling as it does to the moonlight. Thus, DVM disruptions induced by upwelling often interferes with the one generated by moonlight. In contrast, the bottom-intensified along-slope flow generated by downwelling modifies DVM through accumulating zooplankton in the upper intermediate layer occupied by PSW. The baroclinic eddy reverses DVM below the eddy core. We suggest that the zooplankton's response to upwelling, downwelling, and eddy is consistent with adjusting DVM to avoid enhanced water dynamics.

In contrast to many previous studies of the high-Arctic regions, at ~71°N latitude we recorded DVM during the midnight sun. During the ice-free season of the midnight sun 2004, DVM was observed through the PW layer. This DVM is likely limited by the depth of chlorophyll maxima in PSW. In 2005 the midnight sun DVM seemed to be masked by a high acoustic scattering level attributed to warmer and turbid layers observed through PSW.

Our analysis was limited by deficient zooplankton observations conducted around the location of the mooring in 2016 (Figure 1a). Zooplankton observations show that the large copepods *Calanus glacialis* and *Metridia longa* were performing DVM in September. However, a comprehensive analysis of the scattering species comprising DVM is logistically impossible for the long-term deployments in the seasonally ice-covered and remote areas in the high Arctic. This prohibits identification of specific species whose DVM was detected by the 300 kHz ADCP and altered by the different environmental factors including illuminance and water dynamics.



## Acknowledgments

The data used for this research were collected under the ArcticNet framework, project Long-Term Oceanic Observatories in the Canadian Arctic. Camille Lafront at Université Laval did the taxonomic analysis of zooplankton in the selection of plankton net samples made for this study. We gratefully acknowledge the support by the Canada Excellence Research Chair (CERC) and the Canada Research Chairs (CRC) programs. This work is a contribution to the joint Canadian-Danish-Greenland Arctic Science Partnership, Québec-Ocean, and the Canada Research Chair on the response of Arctic marine ecosystems to climate warming. The research was also partly supported by the National Sciences and Engineering Research Council of Canada (IAD: grant RGPIN-2014-03606, JKE: grant RGPIN/435373-2013).

## Data

The ADCP data are available through the Polar Data Catalogue at https://www.polardata.ca/pdcsearch/, CCIN Reference #11653 (Gratton et al., unpublished).

## Authors' contributions

Contributed to conception and design: IAD, VP, GD, DB

Contributed to acquisition of data: JKE, AK, GD, AF, LF

Contributed to analysis and interpretation of data: IAD, VP, AK, GD, SK

Drafted and/or revised the article: IAD, VP, AK, GD

Approved the submitted version for publication: IAD

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

**Figures**


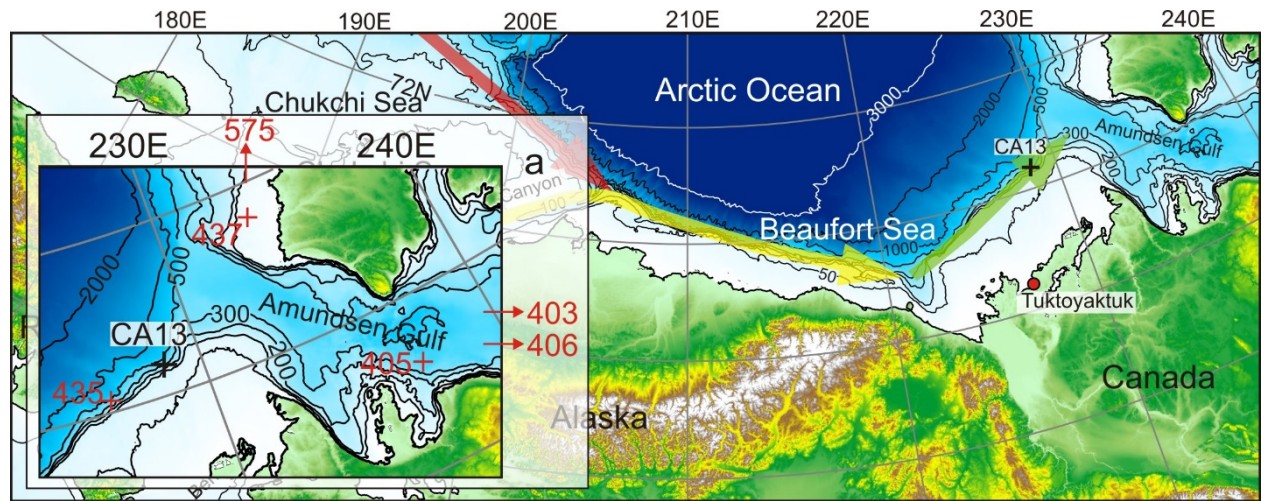

**Figure 1:** Map of the Beaufort Sea with the location of the ArcticNet mooring CA13 (black numbered
cross). Thick red, yellow, and green arrows show circulation associated with the shelfbreak jet over the
Chukchi Sea and western and eastern Beaufort Sea, respectively. (**a**) Inset shows the position of stations
with the zooplankton sampling identified by their station numbers and depicted by red crosses. Thin red
arrows show directions to the zooplankton stations beyond the map limits.





**Figure 2:** Vertical temperature (red), salinity (blue), chlorophyll fluorescence (green) and particulate beam attenuation (black) profiles taken at (**a**) mooring deployment on 9 October 2003, (**b**) on 13 July 2004 and (**c**) at mooring recovery on 4 September 2005. Pink and blue shading and black arrows highlight Pacific Summer Water (PSW), Pacific Winter Water (PWW), and Atlantic Water (AW), respectively, following Dmitrenko et al. (2016).




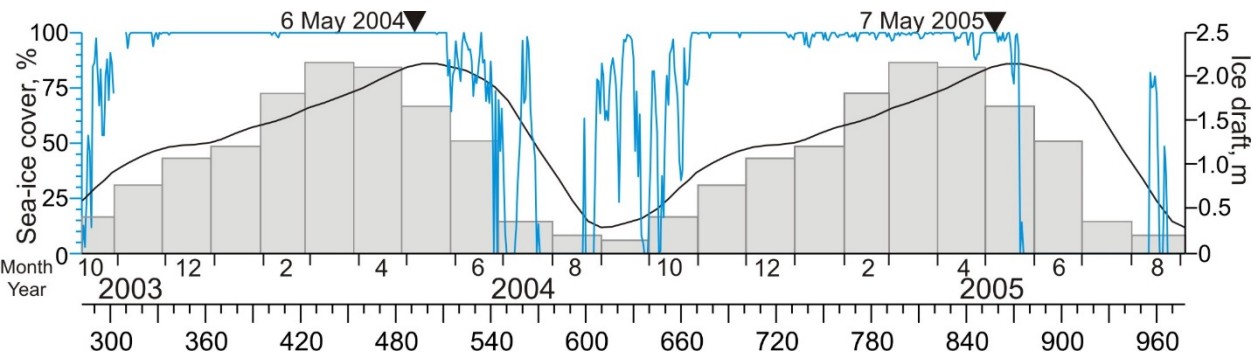


**Figure 3:** Time series of the sea-ice draft (m, gray and black) and concentrations (%, blue). The sea-ice draft annual cycle shown for the first- and multi-year pack ice following Melling et al., 2005 (gray) and Krishfield et al., 2014 (black), respectively. Black triangles at the top identify the time when the RADARSAT satellite images in Figure 5 were acquired.




**Figure 4:** Spatial distribution of sea-ice thickness (cm) over the Canada Basin compiled using gridded sea ice thickness data from ICESat campaigns for (**a**) 24 September – 18 November 2003, (**b**) 17 February – 21 March 2004, (**c**) 3 October – 8 November 2004 and (**d**) 17 February – 24 March 2005 following Kwok et al. (2009). The black cross depicts the mooring position.



**Figure 5:** RADARSAT satellite images taken before sea-ice breakup over the CA13 location northeast
of Cape Bathurst on (**a**) 6 May 2004 and (**c**) 7 May 2005. Yellow stars depict mooring position. Red
rectangular show mooring region enlarged in **b** and **d**. The dark areas are associated with the first-year
pack ice (< 2 m thick). The lighter areas indicate the multi-year pack ice (> 2 m thick).





**Figure 6:** (a) Time series of sea-ice concentrations (blue) and 15-day running mean of total cloud cover
(red). Actograms of (b) modeled under-ice illuminance and (c-g) MVBS at five depth levels: (c) 28 m,
(d) 48 m, (e) 68 m, (f) 88 m, and (g) 108 m. (b) For January-April 2005, the dotted blue line depicts 1
lux threshold estimated for 2.6 m thick ice. Red and blue arrows at the top indicate the polar day and
civil twilight, respectively. Red numbers reference the full moon occurrences, and black horizontal
segments in (a) indicate the mean cloud cover for these periods. Black dashed vertical lines depict
solstices. Red and blue shading highlight the downwelling (D) and upwelling (U) events, respectively,
with their reference numbers on the top. Yellow shading highlights eddies.


**Figure 7:** (**a**) Time series of sea-ice concentrations. Actograms of (**b**) modeled under-ice illuminance and (**c-g**) ADCP-measured vertical velocity (cm s$^{-1}$) at five depth levels: (**c**) 28 m, (**d**) 48 m, (**e**) 68 m, (**f**) 88 m and (**g**) 108 m. Positive/negative values correspond to the upward/downward flow. All other designations are similar to those in Figure 6.




**Figure 8:** Time series of the daily mean relative MVBS (dB) from 28 m to 108 m depth for the astronomic (**a**) midnight and (**b**) noon ±1 h, along-slope (positive northeastward) velocity for depths of (**c**) 28 m and (**d**) 108 m (cm s⁻¹) and water temperatures (°C) for (**c**) 49 m and (**d**) 119 m depth. (**a-b**) Blue lines show the 7-day running mean. Horizontal dotted lines show the 2-year means. Positive/negative values correspond to MVBS gain/loss at 28/108 m depth. (**a**) Gray dashed rectangles depict the full moon occurrence ±6 days. All other designations are similar to those in Figure 6.


**Figure 9**: Enlarged view of the February-March 2004 eddy. (**a**) Zonal and (**b**) meridional current (cm s[-1]) records as functions of depth adopted from Dmitrenko et al. (2018). (**c-d**) actograms of MVBS (dB) for (**c**) 28m and (**d**) 108 m depth.



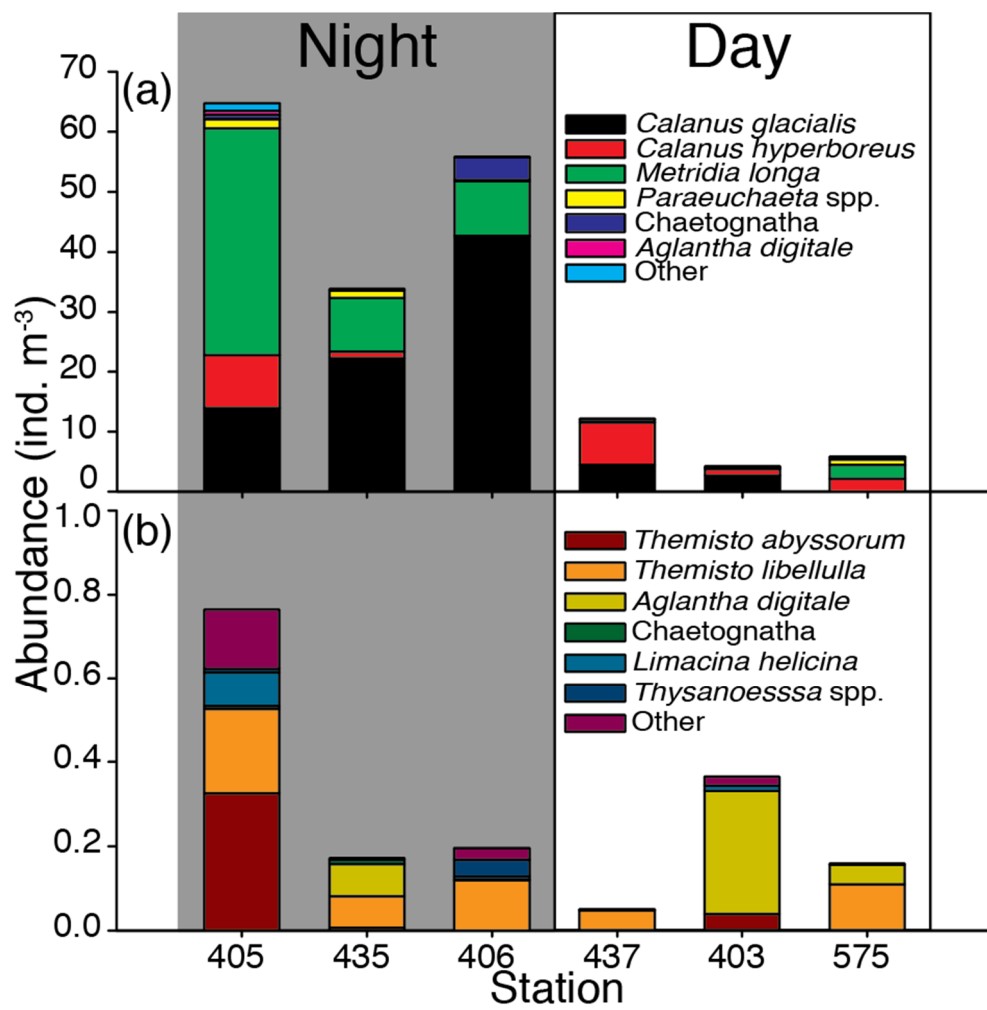

**Figure 10:** Large (**a**) mesozooplankton and (**b**) macrozooplankton numerical composition in the top 90-m layer of the water column during (left) night and (right) day time in the southeastern Beaufort Sea in the first half of September 2016. For station positions see Figure 1a.





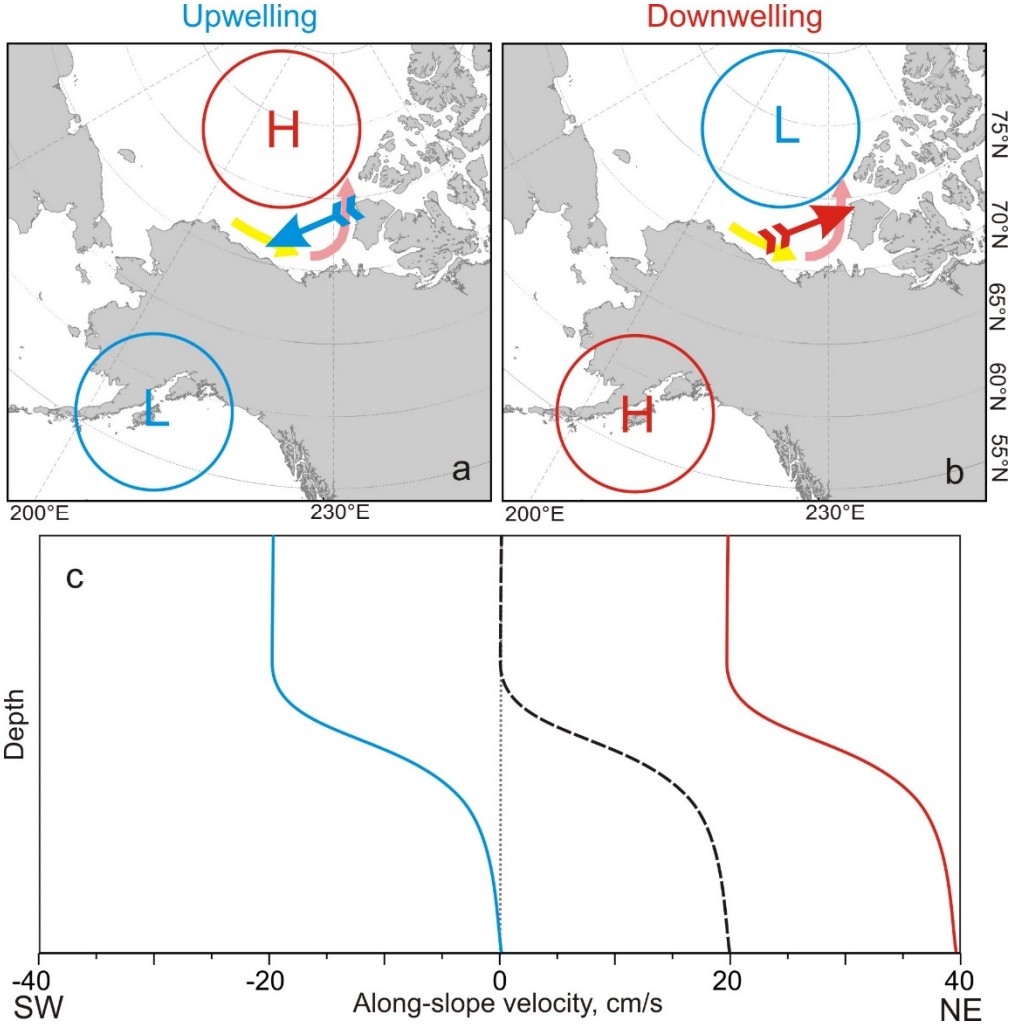

**Figure 11:** Schematic depiction showing atmospheric forcing for (**a**) upwelling and (**b**) downwelling along the eastern Beaufort Sea continental slope adopted from Kirillov et al. (2016). Blue and red arrows indicate geostrophic wind associated with concurrence between atmospheric low and high depicted by blue and red circles, respectively. Yellow and pink arrows show circulation with shelfbreak jet over the western and eastern Beaufort Sea, respectively, intensified by local downwelling. (**c**) Schematic depiction suggesting generation of the surface-intensified (blue curve) and depth-intensified (red curve) along-slope currents as a result of upwelling and downwelling, respectively, superimposed on the hypothetical bottom-intensified shelfbreak current (black dashed curve) following Dmitrenko et al. (2018).