# Peer review of "Sea-ice and water dynamics and moonlight impact the acoustic backscatter diurnal signal over the eastern Beaufort Sea continental slope"

_Ocean Science, 2020_

## Referee Comment (RC1) · Jørgen Berge (Referee) · 22 Jun 2020

Abstract: To which degree will the ADCP "see" suspended particles? Is it really true that suspended particles can mask the backscatter signal from zooplankton? I think this should be included in the discussion as a separate topic

Introduction: Line 45: Zooplankton samples taken 13 years after the mooring was deployed? I would suggest that the part on zooplankton samples is taken out, as it in reality have very little added value. This applies throughout the manuscript

Line 46: "The environmental factors controlling DVM in the seasonally ice-covered Arctic areas,..., remains poorly assessed". There have been numerous studies documenting that it is light that is proximate cue for DVM, both in the Arctic and elsewhere. So if you mean that proximate cues are poorly studied, I would disagree. However, if you are referring to other environmental factor and their effect on DVM, then this is a topic not merely poorly studied in the Arctic, but in general. The effect of upwelling/downwelling is also a novel and important contribution to the understanding of DVM in general! I would recommend rewriting this so that it becomes clearer?

Line 48: To avoid confusion, I would refer to Cohen et al 2020 (table 3.1) with definition of polar night...since you are discussing light levels, the most correct term is actually "civil twilight" which occur during polar twilight when the sun is less than 6 degrees below the horizon. The same applies for the other locations mentioned in the sentences below (need to separate between the definition of types of twilight and polar night periods)

data: Line 83-100: Wallace et al 2010 and later Hobbs et al 2018 used and published a procedure on how to infer ice-cover from an upward-looking ADCP. This would provide a good and in situ data source for ice cover at the mooring site

Lines 101-115: I would argue that this part should be deleted. The samples were collected no less than 13 years after the mooring was deployed, and actually relatively far away from the mooring site. There need to be some seriously strong arguments (that are not provided) for using these samples as a reference point for which scatterers were present 11-13 year earlier. The strength of the manuscript is NOT decreased by omitting these data (quite the opposite, I would argue) - much of the discussion is still valid without referring to zooplankton nets taken in September 2016.

Lines 201-206: Check values...disregarding atmospheric refraction, the polar day and polar night are symmetrical. atmospheric refraction will prolong the observed polar day, but hardly as much as presented here (20 days longer polar day compared to the polar
night).

Discussion Line 438-440: DVM during the polar night is most likely not "diurnal movement of zooplankton towards the surface at dusk", but rather the opposite (movement away for the surface during the short period of increased illumination at around noon). Suttle, but important distinction in order to understand the process of DVM during the polar night (see recent literature on polar night zooplankton and dvm)

Section 7.4: How does this study deviate from previous studies (e.g. from Svalbard) that have aimed at studying the effect of water masses, halo- and pycnoclines, etc?

General comment: I find the results in relation to an absolute threshold of light (lux=1) interesting, but I think the authors have a lot to gain from presenting a more thorough discussion on the importance of light intensity vs rate of change. Most published papers emphasise the rate of change as the important cue. Also, the use of lux is not very common in studies of DVM - is it possible to relate lux to absolute quantas of photons? This would enhance comparison with previous studies.

OSD

---

## Referee Comment (RC2) · Anonymous Referee #2 · 2 Jul 2020

Review of the manuscript "Sea-ice and water dynamics and moonlight impact the acoustic backscatter diurnal signal over the eastern Beaufort Sea continental slope" by Dmitrenko et al.

This paper describes diurnal vertical migration of zooplankton and associated changes in acoustic backscatter intensity measured in the eastern Beaufort Sea in response to daily cycles in light illuminance. Undoubtedly, this study provides a valuable insight into how the polar ecosystem evolves in response to the light cycle and is of interest for polar biologists and oceanographers. However, the manuscript in its present form has

many flaws and therefore needs major revision. In several places, the text is not well-structured and clearly written. The results and discussion are not well separated and include numerous repetitions of the same content in many places. I would recommend the authors revise their text to improve its clarity. Further, I provided my suggestions as to what improvements are needed before this paper may be accepted.

General comments:

The logical link between evaluation of zooplankton communities and DVM at the mooring in 2003-2005 looks weak. This section and further discussion are entirely based on biological samplings collected in 2016 and separated from the time of mooring observations by more than eleven years. This is a considerably long period. I am surprised that the authors push us to believe that nothing happened with zooplankton communities in the Beaufort Sea over this time. It is unlikely that this is true, especially if we all know about substantial changes taking place in almost all components of the Arctic climate system during this time. I'm not claiming that any new biological species appear at the mooring site even if it might be the case, but it's tough to believe, for instance, that the biomass of each identified class remains unchanged. This raises the question how the presented materials correspond to the conditions at the mooring in 2003? Moreover, zooplankton samples in 2016 were collected for one particular month and, thus, cannot be representative for the entire seasonal cycle, making this part even more speculative. The authors are entirely mute about all these uncertainties. I would suggest the authors either improve that part providing more arguments for why we should trust this analysis or completely remove it.

An additional concern arises about sea-ice data. The authors do not really have at hand an appropriate sea ice thickness record to examine its impact on annual changes of zooplankton DVM. Instead, for simulations, they used a mean seasonal cycle of ice thickness that cannot represent natural year-to-year variability at the CA13 mooring. Thus, the comparison of ice impact on the simulated illuminance in 2004 and 2005 looks questionable. The use of the complementary ICESat data set partially tones

down this problem, but does not address it in details because these data are also constrained. In that situation, the option I suggest is to rely on available models like PIOMAS, for example, or other ocean reanalyses data, which assimilate ice concentrations and thicknesses measurements. In that case, the discussion of ice impact on DVM may be more straightforward. It is also not clear to me why the authors used two series (Melling et al. 2005 and Krishfield et al. 2014) of the mean seasonal cycle if none of them are from the mooring site and just illustrate typical evolution of sea ice in the region? What's changed if we remove one of them?

I seriously doubt that the presented analysis of the illuminance due to moonlight was carried out convincingly. Figures 6 and 7, where we should see that impact, are very messy and I personally would not say that they show us this relationship in a clear way. For instance, despite the cloudiness during event#6 being about the same low as for events#3 and #4, we do not see any response in MVBS at 28-m depth. Moreover, in Fig. 7 any pattern due to moon phases is not evident at all.

Specific comments:

L61: "228°38.176'E" Here and throughout the text I suggest using western longitude instead.

L64: "CTD (temperature-salinity-depth)". CTD stands for conductivity-temperature-depth.

L71: "…the RDI reports that the vertical velocity is more accurate than the horizontal velocity by at least a factor of two". Could the authors provide a reference for this statement?

L83-87: Please, specify where these data come from and what algorithm was used for ice data processing? I also wonder why the authors used spatially-averaged sea ice concentrations over a ~200-km rectangle, not just observations at the site of the mooring. The later looks more logical for the purposes of the assessment of light

transmission under specific conditions at the mooring. Could you comment, please?

L94: "...https://rkwok.jpl.nasa.gov/icesat/index.html ". The link provided doesn't exist.

L98-100: "Finally, we used (iv) satellite synthetic aperture radar (SAR) imagery acquired by Canadian RADARSAT over the mooring location 100 before the sea-ice breakup in 2004 and 2005 (Figure 5)". Please, provide a source for these images as well.

L104-105: "...at times as close as possible to local midnight, with 3 other stations sampled during daytime at times close to midday". Specify the largest time difference between the time of profiling and the local midnight and midday.

L127:134: The description of the radiative model is incomplete. It's unclear what was used to calculate illuminance at the top of snow layer. What exactly does the snow thickness series look like? How were ice concentrations and clouds utilized in these calculations? Does this model simulate light distribution in the water layer beneath the sea ice?

L136: "The diurnal signal variation is presented along the vertical axis of the actogram, while the long-term patterns of diurnal behavior ..." The meaning is not clear. Did the authors mean variations during a day-long period? What does a "diurnal behavior" mean?

L157-159: "Overall, satellite data show that during winter-spring 2005 sea-ice thickness over the mooring location exceeded that for 2004 by >1 m suggesting implications for the under-ice illuminance values". I wonder if this conclusion has any impact on the simulated illuminance.

L161: Likely, the "layer" is missing somewhere.

L194: "...diurnal signal variations". As I noted above, this term is unclear. In actograms presented in Figs. 6-7, changes or variations of diurnal signal are shown along the horizontal axis, not the vertical one. Meanwhile, values along the vertical axis indicate

the temporal changes for any specific day, which might be called diurnal cycle or signal, but not its variations. If that is correct, I would suggest the author rephrase this for clarity.

L202-206: Consider merging this paragraph with L45-50 to avoid unnecessary repetition.

L209-210: "Outside of the polar day, the sun illuminance is opposite to MVBS for the sub-surface layer, while at 108 m depth this relationship becomes positive". An awkward sentence because it has meaning only for the diurnal changes, not for the illuminance and MVBS themselves. Please, rephrase.

L211-212: "In spring 2004, the modification of the MVBS diurnal pattern from the beginning of May corresponds to an increase of the midnight under-ice illuminance to >1 lux (Figures 6b and 6c)." I think we need a clarification of what "modification" means here. Is it the vanishing of diurnal pattern? Moreover, I doubt that we can trust this number for the under-ice illuminance if we take into account that sea ice thickness was reproduced by the mean annual cycle.

L216: According to the description, ICESat data were available only for one spring month (March) in 2004 and 2005. Assuming dynamical nature behind the ice thickness anomaly in 2005, you cannot easily extend this conclusion for the entire spring or specifically to May 2005.

L216-218: "In spring 2005, the midnight under-ice illuminance >1 lux was lagging that in 2004 by about one week (blue dotted curve in Figure 6b)". What is the reason of this lag if the under-ice illuminance was simulated using the long-term mean seasonal cycle of sea-ice thickness reported by Melling et al. (2005)?

L218-219: "Note that for winter-spring 2005 the under-ice illuminance, shown in colour in Figure 6b, is overestimated by about a factor of 10 (not shown)". I'm very confused by this statement. If this is true, why do the authors show us an incorrect pattern instead

of reliable values?

L223: "diurnal signal was enhanced". What does the enhancement mean in regards to diurnal signal? Is it the increase of diurnal amplitude or what?

L224-225: "In contrast to the preceding and subsequent periods, no seasonal tendency in the duration of high/low MVBS was observed at this time." The meaning of this is blurry. Could the author formulate this in a more clear way?

L247: "…with the MVBS diurnal rhythm in Figures 5d-5h." Did the authors mean Fig. 6?

L269-270: "During winter, the full moon generates under-ice illuminance up to about 0.1 lux below the sea ice layer with a thickness of around 0.5 m (Figures 3 and 6b)." I cannot understand this. In winter, the sea ice thickness at the mooring site is substantially larger than 1.5 m, not 0.5 m as the authors wrote here. The only months when we see such a thin ice are from July through October, when we see no clear DVM signal.

L273-280: I doubt that this result is well justified. Particularly, I didn't see any clear pattern in MVBS shown in Fig. 6c in response to the moonlight variability. It is assumed that we should see similar inclined straps as we see in Fig.6b, but this is not the case. For some periods (e.g., Nov 2003, Feb-March 2004) we didn't see any response in MVBS at all. Thus, I suggest to find another way to present this analysis.

L298-299: "These deviations, however, can also be attributed to inclinations of the ADCP transducer due to high-velocity currents." Note, that ADCP automatically corrects that inclinations. The reason might be just more dynamical (turbulent) state of the environment associated with larger currents.

L331-332: "moving MVBS upwards." Please, rephrase because MVBS cannot move anywhere.

L337: "It seems that downwelling #3D is entirely dominated by the moon…" Do the authors claim that the Moon impacts downwelling somehow?

L376-377: "It appears that this MVBS anomaly is attributed to the eddy-entrained suspended particles commonly recorded in this area (O'Brien et al., 2011)." If the disruptions of diurnal cycle are attributed to higher concentration of suspended materials in the eddy, why do we see such an unusual pattern in vertical velocity around the core in Fig. 7? I mean very high positive anomalies of the vertical velocity during the day time at 68- and 88-m depths. And a more general question: the authors noted in Discussion that "zooplankton likely avoids enhanced water dynamics" (L539). In that regards, why we do not see vertical migration of both signs in layers above and below the eddy core where we have the strongest currents?

L378: "December/January 2002/2003 ". Check the period.

L403-404: "Our results show that DVM responds to (i) seasonality of the sunlight, (ii) moonlight, and (iii) seasonality of sea-ice cover that attenuates light transmission to the water column." I intuitively agree with that statement, but I should emphasize that the authors partially fail to demonstrate DVM associated with sea ice changes and moonlight variations. See my other comments.

L445-446: "The inter-annual variability in estimated under-ice illuminance is entirely attributed to the sea-ice thickness." How was this conclusion made if the mean annual cycle for sea ice thickness was used to simulate the under-ice illuminance? In Fig. 3, for example, I see no difference in ice draft among 2003/04 and 2004/05.

L446-448: "During the ice season, the mean cloud cover (∼40%) showed insignificant interannual variability (Figure 6a); thus, the cloud cover was not taken into account." This, again, raises the question about what was included in the model to simulate light illuminance.

L550-551: "It appears that upwelling, downwelling, and eddies disrupt DVM by generating a water layer with an enhanced gradient of horizontal velocity." Could the author explain how "the wind-driven barotropic flow generated by upwelling and downwelling wind forcing" (L543) can enhance velocity gradients? If barotropic means

depth-uniform current, it cannot produce vertical gradients. Or do the authors mean lateral gradients?

L594 and Fig. 7: This paragraph is confusing. Why do the events of upwelling and downwelling (e.g., 9U and 10D, and many others) have the same positive sign for the vertical velocity within the entire water layer? If we assume that the response of zooplankton to those events is to avoid layers with enhanced water dynamics (l.601), we should see the opposite direction in zooplankton migration in the case of bottom-intensified downwelling and surface-intensified upwelling. However, following Fig.7, this is not the case. Keeping in mind my previous comment to L376-377, I would say that the presented materials on vertical velocities do not support the author's conclusion at all.

L604-611. Please, see my general comment regarding the analysis of zooplankton.

L621-622: "The ADCP data are available through the Polar Data Catalogue at https://www.polardata.ca/pdcsearch/, CCIN Reference #11653 (Gratton et al., unpublished)." This dataset is not available using the CCIN Reference provided.

Figure 7: I would suggest the author use a different color scheme for this figure to separate positive and negative vertical velocities in a clearer way. For example, using a white color at zero velocity may help.

---

## Author Comment (AC1) · 5 Aug 2020

We highly appreciate helpful comments and suggestions by Prof. Jørgen Berge (Reviewer #1). In the following, the comments by Reviewer #1 are underlined and our responses to the comments are in normal characters. Modifications to the text are shown in quotation marks with bold characters indicating newly added text, and normal characters indicating text that was already present in the previous version. The line numbering is referenced to the marked-up manuscript version.

Referee #1:

1. Abstract: To which degree will the ADCP "see" suspended particles? Is it really true that suspended particles can mask the backscatter signal from zooplankton? I think this should be included in the discussion as a separate topic

There is a large number of papers demonstrating that the sound wave is scattered off the particles, and the backscatter intensity is related to suspended sediment concentration. Over the past few decades, the implication of ADCPs for detection of suspended sediment became commonly accepted (e.g., see overview by Thorne and Hurther, 2014: An overview on the use of backscattered sound for measuring suspended particle size and concentration profiles in non-cohesive inorganic sediment transport studies, doi:10.1016/j.csr.2013.10.017). Nowadays ADCPs (including those operating at 300 KHz) are extensively used for suspended sediment transport monitoring (e.g., Venditti et al., 2016, doi:10.1002/2015WR017348; Dwinovantyo et al., 2017, doi:10.1155/2017/4890421; etc. etc.). Moreover, in 2007, RDI published a technical note providing information on commercially available software packages converting ADCP backscatter data into total suspended sediment concentration as follows: http://www.teledynemarine.com/Documents/Brand%20Support/RD%20INSTRUMENTS/Technical%20Resources/Technical%20Notes/WorkHorse%20-%20ADCP%20Special%20Applications%20and%20Modes/FST017.pdf. Therefore, there is no doubt that ADCPs "see" suspended particles. So, we do not think that discussion on this point is appropriate in the context of our manuscript. Discussion on how the backscatter signal from zooplankton is impacted by suspended particles is already provided in the last paragraph of section 6.1 (lines 590-604). Figures 7d-7f clearly show enhanced MVBS 24 h a day, which is consistent with an acoustic signature of suspended particles. The light attenuation enhanced by suspended particles likely impacts DVM during summer 2005. We agree that term "mask" seems to be inappropriate in this context. DVM can also be impacted by light attenuation generated by enhanced concentration of suspended particles in water column. So, following this comment we changed "*masked*" to "***impacted***" (lines 20, 282, and 590).

2. Introduction: Line 45: Zooplankton samples taken 13 years after the mooring was deployed? I would suggest that the part on zooplankton samples is taken out, as it in reality have very little added value. This applies throughout the manuscript

The part on zooplankton samples was removed, lines 135-149.

3. Line 46: "The environmental factors controlling DVM in the seasonally ice-covered Arctic areas,…, remains poorly assessed". There have been numerous studies documenting that it is light that is

proximate cue for DVM, both in the Arctic and elsewhere. So if you mean that proximate cues are poorly studied, I would disagree. However, if you are referring to other environmental factor and their effect on DVM, then this is a topic not merely poorly studied in the Arctic, but in general. The effect of upwelling/downwelling is also a novel and important contribution to the understanding of DVM in general! I would recommend rewriting this so that it becomes clearer?

Reviewer #1 is correct. To make our statement clearer, we modified this sentence in line 46 as follows: "*The **oceanographic** factors controlling DVM in the seasonally ice-covered Arctic areas... remains poorly assessed*".

4. Line 48: To avoid confusion, I would refer to Cohen et al 2020 (table 3.1) with definition of polar night...since you are discussing light levels, the most correct term is actually "civil twilight" which occur during polar twilight when the sun is less than 6 degrees below the horizon. The same applies for the other locations mentioned in the sentences below (need to separate between the definition of types of twilight and polar night periods)

Following this comment, we modified text in lines 48-54 as follows: "***At t**his latitude **no actual daylight is experienced during short winter daylight hours with the exception of the civil twilight** when solar illumination is still sufficient for the human eye to distinguish terrestrial objects. This geographical position makes our DVM observational site vastly different from those at Svalbard (**astronomical twilight, the Sun is between 12 and 18° below the horizon**, ~80°N; e.g., Grenvald et al., 2016; Darnis et al., 2017), Canada Basin (nautical twilight, the Sun is between 6 and 12° below the horizon, ~77.5°N; La et al., 2018), and Northeast Greenland (nautical twilight, ~74.5°N, Petrusevich et al., 2016)*".

5. Data: Line 83-100: Wallace et al 2010 and later Hobbs et al 2018 used and published a procedure on how to infer ice-cover from an upward-looking ADCP. This would provide a good and in situ data source for ice cover at the mooring site

We estimate the sea-ice cover concentration from AMSR-E. Following comment #8 by Reviewer #2, we changed sea ice concentrations, spatially-averaged over a 200-km rectangle, to that for the single pixel, closest to the mooring position (please see revised text in lines 91-94). The sea-ice thickness, however, remained uncertain. A procedure on how to infer ice-cover thickness from an upward-looking ADCP suggested by Wallace et al. (2010) and later by Hobbs et al. (2018) is based on the algorithm published by Hyatt et al. (2008, doi: 10.1016/j.dsr2.2007.11.004). According to that algorithm, the ADCP bin that samples the sea surface or sea-ice bottom is identified as the bin above the bin with maximum backscatter intensity (Hyatt et al, 2008). For CA13, the velocity and acoustic backscatter data were obtained at 8-m depth intervals, so the last sampled bin #13 was at 4 m depth. This level corresponds to the maximum backscatter intensity (as shown in the figure presented below). Thus, there is no bin above the one with the maximum backscatter intensity, which does not allow to apply this algorithm. To resolve this issue, we followed a recommendation provided by Reviewer #2 (his/her comment #2) to derive sea-ice thickness from the Pan-Arctic Ice Ocean Modeling and Assimilation System (PIOMAS) and Hybrid Coordinate Ocean Model (HYCOM) + Community Ice Code (CICE) coupled ocean and sea ice system. Instead of the mean seasonal cycle, we used grid daily data from PIOMAS and HYCOM+CICE for the grid node closest to the mooring position (please see revised text in lines 103-119).

[Figure]

Figure shows acoustic backscatter data from a 300 kHz upward-looking Workhorse Sentinel ADCP by RDI deployed at CA13

6. Lines 101-115: I would argue that this part should be deleted. The samples were collected no less than 13 years after the mooring was deployed, and actually relatively far away from the mooring site. There need to be some seriously strong arguments (that are not provided) for using these samples as a reference point for which scatterers were present 11-13 year earlier. The strength of the manuscript is NOT decreased by omitting these data (quite the opposite, I would argue) - much of the discussion is still valid without referring to zooplankton nets taken in September 2016.

The part on zooplankton samples was omitted in lines 135-149, 458-466, and 489-515. Figure 1 was modified accordingly, and Figure 10 was removed.

7. Lines 201-206: Check values...disregarding atmospheric refraction, the polar day and polar night are symmetrical. Atmospheric refraction will prolong the observed polar day, but hardly as much as presented here (20 days longer polar day compared to the polar night).

The duration of polar night and polar day (the Sun 24 h a day below and above the horizon, respectively) for CA13 position (71°21.356′N, 228°38.176′E) was taken from https://nrc.canada.ca/en/research-development/products-services/software-applications/sun-calculator/. The NOAA calculator at https://www.esrl.noaa.gov/gmd/grad/solcalc/sunrise.html provides similar results. For the North Pole location, this calculator gives results on the duration of the Polar Night and the midnight sun that are similar to those in your Figure 1.5 caption, page 12, POLAR NIGHT Marine Ecology: Life and Light in the Dead of Night, 2020. Following comment #17 by Reviewer #2, this text was shortened and moved to lines 54-57.

8. Discussion Line 438-440: DVM during the polar night is most likely not "diurnal movement of zooplankton towards the surface at dusk", but rather the opposite (movement away for the surface during the short period of increased illumination at around noon). Suttle, but important distinction in order to understand the process of DVM during the polar night (see recent literature on polar night zooplankton and dvm)

The text in lines 517-525 discusses results on DVM at CA13 with respect to the illuminance threshold. During the polar night, these results clearly show the diurnal movement of zooplankton toward the surface at dusk (higher MVBS highlighted by green color in Figure 7d), and descends back the next morning before the short period of increased illumination at around noon (lower MVBS highlighted by dark blue color in Figure 7d). With respect to the reversed DVM (movement away for the surface during the short period of increased illumination at around noon during the Polar Night), Reviewer #1 likely referenced observations taken during the astronomical twilight at 79-80°N (e.g., Berge et al., 2008; Wallace et al., 2010), where during the civil polar night the Sun is between 12° and 18° below the horizon.  In contrast, we used observations taken during the civil twilight, when the Sun was between 0 and <6° below the horizon. We pointed out this important difference in lines 51-54:  "*This geographical position makes our DVM observational site vastly different from those at Svalbard (**astronomical twilight, the Sun is between 12 and 18° below the horizon**, ~80°N; e.g., Grenvald et al., 2016; Darnis et al., 2017), Canada Basin (**nautical twilight, the Sun is between 6 and 12° below the horizon**, ~77.5°N; La et al., 2018), and Northeast Greenland (**nautical twilight**, ~74.5°N, Petrusevich et al., 2016)*".

9. Section 7.4: How does this study deviate from previous studies (e.g. from Svalbard) that have aimed at studying the effect of water masses, halo- and pycnoclines, etc?

The previous studies from the Svalbard area were focused on the relationship between zooplankton community structure and the local hydrography (e.g. Willis et al., 2008, 2011; Kwasniewski et al., 2012; Berge et al., 2014). In contrast, Section 6.3 (former section 7.4) is focused on the DVM modifications related to water dynamics. Effects of water masses and stratification were not discussed because only limited CTD information is available at CA13.

10. General comment: I find the results in relation to an absolute threshold of light (lux=1) interesting, but I think the authors have a lot to gain from presenting a more thorough discussion on the importance of light intensity vs rate of change. Most published papers emphasise the rate of change as the important cue. Also, the use of lux is not very common in studies of DVM - is it possible to relate lux to absolute quantas of photons? This would enhance comparison with previous studies.

*"Our results on the light threshold are consistent with the preferendum (isolume) hypothesis (e.g., Cohen and Forward, 2009). A variant of the preferendum hypothesis, the absolute intensity threshold hypothesis, suggests that an ascent at sunset is initiated once the light intensity decreases below a particular threshold level and a descent at sunrise occurs when the light intensity increases above the threshold intensity (e.g., Cohen and Forward, 2019). This is in line with our findings on an absolute 0.1-lux threshold of light, which corresponds to the moonlight illuminance at the gibbous moon during clear sky (Gaston et al., 2014)".* We added this statement in lines 525-530. We introduced an artificial visual boundary on the illuminance colour scheme at 1 lux (gray to orange), which corresponds to illuminance during the deep twilight – lines 177-179. For the sunlight, 1 lux corresponds to about 0.019 micromoles photons per square meter per second ($\mu$mole photons $m^{-2}\,s^{-1}$). Following this comment, we introduced this unit for the color scale of the under-ice illuminance in Figures 3c-3e.

---

## Author Comment (AC2) · 5 Aug 2020

We highly appreciate helpful comments and suggestions by Reviewer #2. In the following, the comments by Reviewer #2 are underlined and our responses to the comments are in normal characters. Modifications to the text are shown in quotation marks with bold characters indicating newly added text, and normal characters indicating text that was already present in the previous version. The line numbering is referenced to the marked-up manuscript version.

Anonymous Referee #2:

This paper describes diurnal vertical migration of zooplankton and associated changes in acoustic backscatter intensity measured in the eastern Beaufort Sea in response to daily cycles in light illuminance. Undoubtedly, this study provides a valuable insight into how the polar ecosystem evolves in response to the light cycle and is of interest for polar biologists and oceanographers. However, the manuscript in its present form has many flaws and therefore needs major revision. In several places, the text is not well structured and clearly written. The results and discussion are not well separated and include numerous repetitions of the same content in many places. I would recommend the authors revise their text to improve its clarity. Further, I provided my suggestions as to what improvements are needed before this paper may be accepted.

General comments:

1. The logical link between evaluation of zooplankton communities and DVM at the mooring in 2003-2005 looks weak. This section and further discussion are entirely based on biological samplings collected in 2016 and separated from the time of mooring observations by more than eleven years. This is a considerably long period. I am surprised that the authors push us to believe that nothing happened with zooplankton communities in the Beaufort Sea over this time. It is unlikely that this is true, especially if we all know about substantial changes taking place in almost all components of the Arctic climate system during this time. I'm not claiming that any new biological species appear at the mooring site even if it might be the case, but it's tough to believe, for instance, that the biomass of each identified class remains unchanged. This raises the question how the presented materials correspond to the conditions at the mooring in 2003? Moreover, zooplankton samples in 2016 were collected for one particular month and, thus, cannot be representative for the entire seasonal cycle, making this part even more speculative. The authors are entirely mute about all these uncertainties. I would suggest the authors either improve that part providing more arguments for why we should trust this analysis or completely remove it.

This section in lines 135-149, 458-466, and 489-515 was completely omitted, also following recommendations #2 and #6 by Reviewer #1. Figure 1 was modified accordingly, and Figure 10 was removed.

2. An additional concern arises about sea-ice data. The authors do not really have at hand an appropriate sea ice thickness record to examine its impact on annual changes of zooplankton DVM. Instead, for simulations, they used a mean seasonal cycle of ice thickness that cannot represent natural year-to-year variability at the CA13 mooring. Thus, the comparison of ice impact on the simulated illuminance in 2004 and 2005 looks questionable. The use of the complementary ICESat data set partially

tones down this problem, but does not address it in details because these data are also constrained. In that situation, the option I suggest is to rely on available models like PIOMAS, for example, or other ocean reanalyses data, which assimilate ice concentrations and thicknesses measurements. In that case, the discussion of ice impact on DVM may be more straightforward.

Following this comment, the sea-ice description in the manuscript was extensively improved and extended. As part of responding to this comment, we completely removed a mean seasonal cycle of ice thickness by Melling et al. (2005) and Krishfield et al. (2014) from our simulations of under-ice illuminance. Following recommendations by Reviewer #2, we use the PIOMAS daily data on ice thickness at mooring position from http://psc.apl.uw.edu/research/projects/arctic-sea-ice-volume-anomaly/data/model_grid. New text was introduced in lines 103-111 as follows: "***(ii) For sea-ice thickness, we used grid daily data from the Pan-Arctic Ice Ocean Modeling and Assimilation System (PIOMAS, http://psc.apl.uw.edu/research/projects/arctic-sea-ice-volume-anomaly/data/) developed at the Polar Science Center, University of Washington. PIOMAS is a coupled ocean and sea ice model that assimilates daily sea ice concentration and sea surface temperature satellite products (Zhang and Rothrock, 2003). We used data from the grid node at 71.3°N, 226.7°E closest to the mooring position. Schweiger et al. (2011) reported, that PIOMAS spatial thickness patterns agree well with Ice, Cloud, and land Elevation Satellite (ICESat) thickness estimates (also used in this study) with pattern correlations of above 0.8. However, PIOMAS tends to overestimate thicknesses for the thin ice area around the Beaufort Sea, and underestimate the thick ice area around northern Greenland and the Canadian Archipelago (Wang et al., 2016). The overall differences between PIOMAS and ICESat is -15% or -0.31 m (Wang et al., 2016)***". Panel b in Figure 3, showing time series of the PIOMAS derived sea-ice thickness, was modified accordingly. We also modified text in lines 167-168, describing sea-ice data used for simulating under-ice illuminance: "*For computing under-ice illumination in Figures **3e...**, **we use daily PIOMAS... data ... on the simulated sea-ice thickness...***". As an alternative source of sea-ice thickness data, we used simulations based on the Hybrid Coordinate Ocean Model (HYCOM) + Community Ice Code (CICE) coupled ocean and sea ice system developed at the Danish Meteorological Institute, lines 111-119: "***As an alternative source of sea-ice thickness data, we used (iii) simulations based on the Hybrid Coordinate Ocean Model (HYCOM, v2.2.98; e.g. Chassignet et al., 2007) + Community Ice Code (CICE, v4.0; e.g. Hunke, 2001) coupled ocean and sea ice system, developed at the Danish Meteorological Institute (DMI, Madsen et al., 2016). The horizontal resolution is ~10 km. The model domain covers the Arctic Ocean and the Atlantic Ocean down to ~20°S. Madsen et al. (2016) reported that the simulated sea-ice thickness distribution near the Canadian Arctic Archipelago and the northern coast of Greenland is consistent with CryoSat-2 satellite measurements and the NASA Operation IceBridge airborne observations. Simulated sea-ice thickness, shown in Figure 3b, was derived for the grid node closest to the mooring position. Spatial distributions of sea-ice thickness (Figures 4, 6e, and 6f) were acquired from http://ocean.dmi.dk/arctic/icethickness/thk.uk.php***". Results on illuminance simulations using HYCOM + CICE are presented in new Figure 3e.

3. It is also not clear to me why the authors used two series (Melling et al. 2005 and Krishfield et al. 2014) of the mean seasonal cycle if none of them are from the mooring site and just illustrate typical evolution of sea ice in the region? What's changed if we remove one of them?

Both data sets were eliminated from our analysis. They were replaced with the PIOMAS daily data on ice thickness, as recommended by Reviewer #2, and with the HYCOM + CICE data.

4. I seriously doubt that the presented analysis of the illuminance due to moonlight was carried out convincingly. Figures 6 and 7, where we should see that impact, are very messy and I personally would not say that they show us this relationship in a clear way. For instance, despite the cloudiness during event#6 being about the same low as for events #3 and #4, we do not see any response in MVBS at 28-m depth. Moreover, in Fig. 7 any pattern due to moon phases is not evident at all.

We partly agreed with Reviewer #2. There is no disruption of the DVM signatures in vertical velocity, which can be attributed to the moonlight (Figure 8). We added the corresponding statement in lines 609-610: "***Moreover, under-ice vertical velocity data does not show DVM disruptions during full moon phases (Figure 8)***". We also agree that Figure 7 is messy because the water dynamics and full moon events are often overlaid. For example, the full moon event #6 in February-March 2004, mentioned by Reviewer #2, overlaid with eddy, passing the mooring position from 25 February to 4 March (Figure 10). The uncertain cloud cover also introduces an additional complication. However, there are several full moon events, when the DVM disruptions are obvious. For example, in lines 342-347 we focused on the full moon event #1 in September-October 2004, which clearly shows the downward displacement of the acoustic backscatters in response to the moonlight (Figures 7b, c, g, and f). The full moon event #3 in December 2003 (Figure 7b) also clearly shows the downward displacement of the acoustic backscatters from 24 m (Figure 7c) to 68 m (Figure 7e). We introduced a new piece of text in the beginning of section 6.2, discussing moonlight impact on DVM (lines 604-610): "***In general, interpretation of the DVM modifications due to the moonlight is not straightforward. The dark-time MVBS in Figure 7c shows cumulative effect of sea-ice, cloud cover, water dynamics and moonlight. Individual events are often overlaid, and uncertainty in cloud cover also introduces an additional complication. Furthermore, during February-March, the moonlight below sea-ice is strongly attenuated (2004) or completely absorbed by sea-ice (2005) - Figure 3e. Moreover, under-ice vertical velocity data does not show DVM disruptions during full moon phases (Figure 8)***". We also modified the sentence in lines 615-616 as follows: "***While our results on the moon's modifications of DVM are not entirely conclusive, they are** consistent with those previously reported for the Arctic and sub-Arctic regions*". In this context, we also modified our sentence in lines 482-483: "*Our results show that DVM responds to (i) seasonality of the sunlight, (ii) moonlight, and (iii) seasonality of sea-ice cover that attenuates light transmission to the water column, **and to a lesser extent to (iii) moonlight***". Finally, sea-ice thicknesses from HYCOM + CICE helped to reveal that the moonlight during full moon events in February-March 2005 was completely terminated by the ice cover exceeding 2.5 m thick. We added this statement in lines 338-339: "...***3 events in February-March (#7 in 2004 and #6 and 7 in 2005) show complete cessation of the moonlight transmittance through sea-ice exceeding 2.5 m thick (Figure 7b)***" and lines 327-328: "***Sea-ice strongly attenuates moonlight. Once sea-ice thickness exceeded ~2.5 in April 2004 and February 2005, the moonlight transmittance through sea-ice is completely terminated (Figures 3b and 3e)***".

Specific comments:

5. L61: "228° 38.176'E" Here and throughout the text I suggest using western longitude instead.

Changed as requested, line 66, Figures 1 and 10.

6. L64: "CTD (temperature-salinity-depth)". CTD stands for conductivity-temperature-depth.

Thank you. Corrected in line 69.

7. L71: "…the RDI reports that the vertical velocity is more accurate than the horizontal velocity by at least a factor of two". Could the authors provide a reference for this statement?

Thank you. This reference to RDI is wrong. We improved this statement in lines 75-77 as follows: "*The accuracy of the ADCP vertical velocity measurements is not validated; however, **for the 600 kHz RDI ADCP, Wood and Gartner (2010)** reported that the vertical velocity is more accurate than the horizontal velocity by at least a factor of two*".

8. L83-87: Please, specify where these data come from and what algorithm was used for ice data processing? I also wonder why the authors used spatially-averaged sea ice concentrations over a 200-km rectangle, not just observations at the site of the mooring. The later looks more logical for the purposes of the assessment of light transmission under specific conditions at the mooring. Could you comment, please?

The data came from https://icdc.cen.uni-hamburg.de/en/seaiceconcentration-asi-amsre.html. They have been computed by applying the ARTIST Sea Ice (ASI) algorithm to brightness temperatures measured with the 89 GHz AMSR-E channels (Spreen et al., 2008). We agree with Reviewer #2 that the sea-ice concentrations derived for the site of the mooring are more appropriate in this context. So, we replaced spatially-averaged sea ice concentrations with those from the pixel closest to the mooring position (see modified Figures 3b, 7a, and 8a). Under-ice illuminance in Figures 7b and 8b was recomputed accordingly. Responding this comment by Reviewer #2, we modified text in lines 91-94 as follows: "***They have been computed by applying the ARTIST Sea Ice (ASI) algorithm to brightness temperatures measured with the 89 GHz AMSR-E channels, and are available through https://icdc.cen.uni-hamburg.de/en/seaiceconcentration-asi-amsre.html. The ASI algorithm is described in Spreen et al. (2008).*** *The spatial grid resolution for ice concentration is 6.25 km,* **and we used data from the pixel, closest to the mooring position**".

9. L94: ": : :https://rkwok.jpl.nasa.gov/icesat/index.html ". The link provided doesn't exist.

Thank you. Yes, this link certainly disappeared. We modified this sentence in lines 120-122 as follows: "We also used data on sea-ice thickness from ICESat obtained **from the NASA National Snow and Ice Data Center (Yi and Zwally, 2009)**".

10. L98-100: "Finally, we used (iv) satellite synthetic aperture radar (SAR) imagery acquired by Canadian RADARSAT over the mooring location before the sea-ice breakup in 2004 and 2005 (Figure 5)". Please, provide a source for these images as well.

Responding this comment, we introduced new sentence in lines 127-129 as follows: "**RADARSAT data were acquired through the Government of Canada's Earth Observation Data Management System (https://www.eodms-sgdot.nrcan-rncan.gc.ca**) ".

11. L104-105: "…at times as close as possible to local midnight, with 3 other stations sampled during daytime at times close to midday". Specify the largest time difference between the time of profiling and the local midnight and midday.

This text was omitted following comments #1 by Reviewer #2 and #2 and 6 by Reviewer #1.

12. L127:134: The description of the radiative model is incomplete. It's unclear what was used to calculate illuminance at the top of snow layer. What exactly does the snow thickness series look like?

How were ice concentrations and clouds utilized in these calculations? Does this model simulate light distribution in the water layer beneath the sea ice?

This model does not simulate light distribution in the water layer beneath the sea ice. In the original version, we used snow thickness on the top of the ice gradually increasing from zero at freeze-up to a typical 15-cm thick wind-packed snow in late winter (Melling et al., 2005). In the revised version, however, we derived snow data from AMSR-E/Aqua observations, lines 130-134: "***Snow depth over sea-ice, derived from AMSR-E/Aqua, was obtained from NSIDC (Cavalieri et al., 2014). The 12.5 km snow depth is provided as a 5-day running average. It is generated using the AMSR-E snow-depth-on-sea-ice algorithm based on the spectral gradient ratio of the 18.7 GHz and 36.5 GHz vertical polarization channels (Markus and Cavalieri, 1998). As of the AMSR-E sea-ice concentrations, for generating time series of the snow depth over sea-ice (Figure 3a) we used data from the pixel closest to the mooring position***". We also added this information in lines 169-170: "***The snow thickness on the top of the ice was taken from AMSR-E/Aqua observations***". We added new paned (a) in Figure 3 showing time series of snow thickness on the top of the ice. We also modified text in lines 167-168, describing sea-ice data used for simulating under-ice illuminance: "*For computing under-ice illumination in Figures **3d** and **3e**, **we used PIOMAS and HYCOM+CICE data on the simulated sea-ice thickness, respectively**"*. We accounted for the sea-ice if its concentration exceeds 90%. We added this statement in lines 170-171: "***We accounted for the sea-ice and snow cover if the sea-ice concentration exceeds 90%***". Finally, we specifically pointed out that the cloud cover data were not used for simulating under-ice illuminance, lines 171-172: "***Cloud cover information is not utilized by this model due to high uncertainty of the cloud cover data (Liu and Key, 2016)***".

13. L136: "The diurnal signal variation is presented along the vertical axis of the actogram, while the long-term patterns of diurnal behavior…" The meaning is not clear. Did the authors mean variations during a day-long period? What does a "diurnal behavior" mean?

Yes, the "*diurnal behavior*" means variations during a day-long period. We modified this sentence in lines 175-176 as follows: "***Variations during a day-long period are*** *presented along the vertical axis of the actogram…*".

14. L157-159: "Overall, satellite data show that during winter-spring 2005 sea-ice thickness over the mooring location exceeded that for 2004 by >1 m suggesting implications for the under-ice illuminance values". I wonder if this conclusion has any impact on the simulated illuminance.

First, this statement was strengthened involving PIOMAS and HYCOM+CICE data as recommended by Reviewer #2. Second, we compared PIOMAS to HYCOM+CICE and satellite information and revealed that the HYCOM+CICE simulations are better in reproducing sea-ice thickness in winter 2005. This is described in the new paragraph introduced in lines 198-213 alone with impact of the difference between sea-ice thickness in 2004 and 2005 on the simulated illuminance: "***The satellite information on sea-ice thickness, however, is not consistent with PIOMAS. For February-March 2004 and 2005, PIOMAS provides estimates of sea-ice thickness at mooring position of 1.87 m and 2.28 m, respectively (Figure 3b). In contrast, for the same time period, ICESat provides 1.5-1.4 m and 2.4-2.5 m, respectively (Figures 4c and 4d). This discrepancy is in line with the conclusions by Wang et al. (2016) that PIOMAS overestimates thicknesses for the thin ice area around the Beaufort Sea and underestimates the thick ice area around the northern Greenland and the Canadian Arctic Archipelago. For winter-spring 2003-04, PIOMAS data agree relatively well with HYCOM+CICE data (Figure 3b). For January-May 2005,***

*however, the discrepancy between PIOMAS and HYCOM+CICE increases from ~0.5 m on 1 January 2005 to ~1.3 m on 22 May 2005 (Figure 3b). During winter-spring 2005, spatial distribution of sea-ice thickness, derived from HYCOM+CICE simulations, shows the on-slope displacement of the multi-year pack ice from the Greenland and Ellesmere Island shelves (Figure 4), which is also revealed from the satellite observations (Figures 5 and 6a-6d). For winter-spring 2005, the HYCOM+CICE data on the multi-year pack ice >2 m thick over the mooring position are in line with detecting multi-year ice on the RADARSAT satellite imagery acquired before sea-ice breakup in May 2005 (Figure 6).* Overall, **the HYCOM+CICE simulations and satellite data suggest show that** *during winter-spring 2005 sea-ice thickness over the mooring location exceeded that for 2004 by* **~1 m with important implication** *for the under-ice illuminance values* **as evident from actograms of under-ice illuminance in Figures 3d and 3e. In what follows, we use under-ice illuminance derived using the HYCOM+CICE simulations"**.

15. L161: Likely, the "layer" is missing somewhere.

Fixed in line 215.

16. L194: "...diurnal signal variations". As I noted above, this term is unclear. In actograms presented in Figs. 6-7, changes or variations of diurnal signal are shown along the horizontal axis, not the vertical one. Meanwhile, values along the vertical axis indicate the temporal changes for any specific day, which might be called diurnal cycle or signal, but not its variations. If that is correct, I would suggest the author rephrase this for clarity.

We modified the sentence in lines 248-249 as follows: "*These actograms reveal a rhythm of activity with diurnal* **cycle** *seen in the vertical axis of an actogram. The 2-year long variability of the diurnal* **cycle** *is observed along the horizontal axis*".

17. L202-206: Consider merging this paragraph with L45-50 to avoid unnecessary repetition.

Thank you. This text was shortened and moved to lines 54-57: "*Civil twilight is observed at the CA13 latitude from 19 November to 21 January. For the winter solstices (22 December), the civil twilight lasts for about 3 h. The polar day (or the midnight sun, the Sun is above the horizon for the entire 24 hours) lasts at the CA13 latitude from 10 May to 1 August*".

18. L209-210: "Outside of the polar day, the sun illuminance is opposite to MVBS for the sub-surface layer, while at 108 m depth this relationship becomes positive". An awkward sentence because it has meaning only for the diurnal changes, not for the illuminance and MVBS themselves. Please, rephrase.

We modified this sentence in lines 263-264 as follows: "*Outside of the polar day,* **the diurnal changes in** *the sun illuminance* **are** *opposite to MVBS for the sub-surface layer, while at 108 m depth this relationship becomes positive*".

19. L211-212: "In spring 2004, the modification of the MVBS diurnal pattern from the beginning of May corresponds to an increase of the midnight under-ice illuminance to >1 lux (Figures 6b and 6c)." I think we need a clarification of what "modification" means here. Is it the vanishing of diurnal pattern? Moreover, I doubt that we can trust this number for the under-ice illuminance if we take into account that sea ice thickness was reproduced by the mean annual cycle.

Yes, we mean the vanishing of diurnal pattern. To clarify this, we slightly modified the sentence in line 265 as follows: "*In spring 2004, the* **vanishing** *of the MVBS diurnal pattern…*". The under-ice illuminance

was recomputed using the PIOMAS and HYCOM+CICE sea-ice thickness (new Figures 3d and 3e, respectively), and snow depth from AMSR-E/Aqua (lines 130-134) following your comments #2 and #12, respectively.

20. L216: According to the description, ICESat data were available only for one spring month (March) in 2004 and 2005. Assuming dynamical nature behind the ice thickness anomaly in 2005, you cannot easily extend this conclusion for the entire spring or specifically to May 2005.

To extend this conclusion for the entire spring 2005, or specifically to May 2005, we used RADARSAT satellite imagery taken before sea-ice breakup on 7 May 2005 (Figures 6c and 6d). This is pointed out in lines 192-197: "*The on-slope displacement of the multi-year pack ice from the Greenland and Ellesmere Island shelves **was observed during winter 2005**. This is evident from the sea-ice thickness ICESat data showing a west-southward expansion of the Greenland pack in February - March 2005 (Figure 5d). This is in line with detecting multi-year ice on the RADARSAT satellite imagery acquired over the mooring position **in May 2005** (Figure 6). The lighter areas in Figures 6c and 6d indicate the multi-year pack ice expanded over the mooring position before the sea-ice breakup in May 2005*". However, this assumption is not consistent with PIOMAS sea-ice thickness data (Figure 3b). The possible explanation is that "...***PIOMAS tends to underestimate the thick ice area around the northern Greenland and the Canadian Arctic Archipelago (Wang et al., 2016)***" – lines 109-110. In our case, for February-March 2004 and 2005, PIOMAS provides estimates of the sea-ice thickness at mooring position of 1.87 m and 2.28 m, respectively (Figure 3b). In contrast, for the same time period, ICESat gives 1.5-1.4 m and 2.4-2.5 m, respectively (Figures 4c and 4d). For justifying our conclusions on sea-ice ice thickness anomaly in 2005, we used HYCOM+CICE simulations (new Figure 4 and new panels e and f in Figure 6), and lines 205-213: "***During winter-spring 2005, spatial distribution of sea-ice thickness, derived from HYCOM+CICE simulations, shows the on-slope displacement of the multi-year pack ice from the Greenland and Ellesmere Island shelves (Figure 4) similar to that revealed from the satellite information (Figures 5 and 6a-6d). For winter-spring 2005, the HYCOM+CICE data on the multi-year pack ice >2 m thick over the mooring position are in line with detecting multi-year ice on the RADARSAT satellite imagery acquired before sea-ice breakup in May 2005 (Figure 6). Overall, the HYCOM+CICE simulations and** satellite data **suggest** that during winter-spring 2005 sea-ice thickness over the mooring location exceeded that for 2004 by **~1 m with important implication for the under-ice illuminance values as evident from actograms of under-ice illuminance in Figures 3d and 3e. In what follows, we use under-ice illuminance derived using the HYCOM+CICE simulations*".

21. L216-218: "In spring 2005, the midnight under-ice illuminance >1 lux was lagging that in 2004 by about one week (blue dotted curve in Figure 6b)". What is the reason of this lag if the under-ice illuminance was simulated using the long-term mean seasonal cycle of sea-ice thickness reported by Melling et al. (2005)?

ICESat and RADARSAT data show that during winter 2005 sea-ice was ticker compared to winter 2004 by ~1 m. In the original version of the manuscript, we simulated the difference between winter 2004 and 2005 by artificially increasing the sea-ice thickness since 1 January 2005 by 1 m. In the revised version of the manuscript, we computed the under-ice illuminance using the snow cover and sea-ice thickness provided by the satellite data (Figure 3a) and HYCOM+CICE simulations (Figure 3b), respectively. Newly generated Figures 3e, 7b and 8b show the difference in simulated 1-lux and 0.1-lux thresholds for

winters 2004 and 2005. As a result of the thicker ice in April-May 2005, "...*the midnight under-ice illuminance >1 lux was lagging that in 2004 by about one week (Figure **7**b)*", lines 271-272.

22. L218-219: "Note that for winter-spring 2005 the under-ice illuminance, shown in colour in Figure 6b, is overestimated by about a factor of 10 (not shown)". I'm very confused by this statement. If this is true, why do the authors show us an incorrect pattern instead of reliable values?

The values of illuminance shown in the original Figures 6b and 7b were obtained based on the seasonal cycle reported by Krishfield et al. (2014) – lines 158-159 in the manuscript original version. The satellite information (both, ICESat and RADARSAT), however, provides an evidence, that the sea-ice thickness in winter 2005 exceeded that for winter 2004 by ~1 m. In the original version of the manuscript, we added this difference to the Krishfield's seasonal cycle since 1 January 2005 to reveal the response of the 1-lux threshold to the ticker ice in 2005. In the revised version, we replaced the Krishfield's seasonal cycle with the PIOMAS and HYCOM+CICE sea-ice thickness as recommended by Reviewer #2. After the PIOMAS data were verified against the satellite data, we chose the HYCOM+CICE simulations. More details explaining this choice are provided in the newly introduced paragraph in lines 198-213. Revised Figures 7b and 8b now show "reliable" values for the under-ice illuminance, which were simulated using HYCOM+CICE data.

23. L223: "diurnal signal was enhanced". What does the enhancement mean in regards to diurnal signal? Is it the increase of diurnal amplitude or what?

Yes, we mean the increase of the MVBS diurnal amplitude. We modified this sentence in lines 277-278 for clarity: "*From **about** 1 April to 10 July 2004, **the diurnal amplitude** of MVBS signal was enhanced at the **68-108 m depth layer due to the MVBS values lowered from ~ -61 to -66 dB** (Figures **7**e-**7**g) **during the astronomic midnight ±3 h**".

24. L224-225: "In contrast to the preceding and subsequent periods, no seasonal tendency in the duration of high/low MVBS was observed at this time." The meaning of this is blurry. Could the author formulate this in a more clear way?

We mean that we did not observe seasonal modulation of the MVBS diurnal cycle. To clarify it, we modified this sentence in lines 278-280 as follows: "*In contrast to the preceding and subsequent periods, **no seasonal modulation of the MVBS diurnal cycle** was observed at this time*".

25. L247: "...with the MVBS diurnal rhythm in Figures 5d-5h." Did the authors mean Fig. 6?

Yes, thank you. Fixed in line 302.

26. L269-270: "During winter, the full moon generates under-ice illuminance up to about 0.1 lux below the sea ice layer with a thickness of around 0.5 m (Figures 3 and 6b)." I cannot understand this. In winter, the sea ice thickness at the mooring site is substantially larger than 1.5 m, not 0.5 m as the authors wrote here. The only months when we see such a thin ice are from July through October, when we see no clear DVM signal.

Thank you. You are totally right. We modified this sentence in lines 324-326 as follows: "*During **mid-winter (end of December)**, the full moon generates under-ice illuminance up to about **0.001** lux below the sea-ice layer with a thickness of around **1** m **and ~20 cm snow depth over sea-ice** (Figures **3b** and **7b)**". We also added two new sentences following this statement (lines 326-328): "***In contrast, for the***

*open water conditions, the full moon generates illuminance exceeding 0.1 lux (Figure 3c). Sea-ice strongly attenuates moonlight. Once sea-ice thickness exceeded ~2.5 m in April 2004 and February 2005, the moonlight transmittance through sea-ice is completely terminated (Figures 3b and 3e)"*.

27. L273-280: I doubt that this result is well justified. Particularly, I didn't see any clear pattern in MVBS shown in Fig. 6c in response to the moonlight variability. It is assumed that we should see similar inclined straps as we see in Fig.6b, but this is not the case. For some periods (e.g., Nov 2003, Feb-March 2004) we didn't see any response in MVBS at all. Thus, I suggest to find another way to present this analysis.

We partly agreed with Reviewer #2. There is no disruption of the DVM signatures in vertical velocity, which can be attributed to the moonlight (Figure 8). We added the corresponding statement in lines 609-610: "*Moreover, under-ice vertical velocity data does not show DVM disruptions during full moon phases (Figure 8)*". We also agree that Figure 7 is messy because the water dynamics and full moon events are often overlaid. For example, the full moon event #6 in February-March 2004, mentioned by Reviewer #2, overlapped with eddy, passing mooring position from 25 February to 4 March (Figure 10). The uncertain cloud cover also introduces an additional complication. However, there are several full moon events, when the DVM disruptions are obvious. For example, in lines 342-347 we focused on the full moon event #1 in September-October 2004, which clearly shows the downward displacement of the acoustic backscatters in response to the moonlight (Figures 7b, c, g, and f). The full moon event #3 in December 2003 (Figure 7b) also clearly shows the downward displacement of the acoustic backscatters from 24 m (Figure 7c) to 68 m (Figure 7e). Responding this comment, we introduced a new text at the beginning of section 6.2, discussing moonlight impact on DVM (lines 606-610): "*In general, interpretation of the DVM modifications due to the moonlight is not straightforward. The dark-time MVBS in Figure 7c shows cumulative effect of sea-ice, cloud cover, water dynamics and moonlight. Individual events are often overlaid, and uncertainty in cloud cover also introduces an additional complication. Furthermore, during February-March, the moonlight below sea-ice is strongly attenuated (2004) or completely absorbed by sea-ice (2005) - Figure 3e. Moreover, under-ice vertical velocity data does not show DVM disruptions during full moon phases (Figure 8)*". We also modified the sentence in lines 615-616 as follows: "*While our results on the moon's modifications of DVM are not entirely conclusive, they are* consistent with those previously reported for the Arctic and sub-Arctic regions*". In this context, we also modified our sentence in lines 482-483: "*Our results show that DVM responds to (i) seasonality of the sunlight, (ii) moonlight, and (iii) seasonality of sea-ice cover that attenuates light transmission to the water column, and to a lesser extent to (iii) moonlight*". Finally, sea-ice thicknesses from HYCOM + CICE helped to reveal that the moonlight during full moon events in February-March 2005 was completely terminated by the ice cover exceeding 2.5 m thick. We added this statement in lines 338-339: "*...3 events in February-March (#7 in 2004 and #6 and 7 in 2005) show complete cessation of the moonlight transmittance through sea-ice exceeding 2.5 m thick (Figure 7b)"* and lines 327-328: "*Sea-ice strongly attenuates moonlight. Once sea-ice thickness exceeded ~2.5 m in April 2004 and February 2005, the moonlight transmittance through sea-ice is completely terminated (Figures 3b and 3e)*".

28. L298-299: "These deviations, however, can also be attributed to inclinations of the ADCP transducer due to high-velocity currents." Note, that ADCP automatically corrects that inclinations. The reason might be just more dynamical (turbulent) state of the environment associated with larger currents.

Thank you. Responding this comment, we introduced a new paragraph in lines 355-363 as follows: "***The vertical velocity component is very sensitive to spatial inhomogeneity of the flow field and errors in the ADCP tilt angle, introducing errors and significant contamination to the measured vertical velocity component (Ott, 2005). This is consistent with contamination of the vertical velocity data observed during upwellings, downwellings and eddy passing (Figures 8c-8g). Deviations of vertical velocity diurnal pattern can also be attributed to more dynamical (turbulent) state of the environment associated with high-velocity currents.*** *In what follows, we are only interested in the vertical velocity estimates, which are sensitive to the MVBS diurnal cycling.* ***The contaminated vertical velocity data cannot be used for interpretation DVM modifications imposed by upwellings, downwellings, and eddies. Therefore, our analysis on impact of the major energetic events on DVM is entirely based on vertical redistribution of the acoustic backscatters (Figures 7c-7g, 9a and 9b)***".

29. L331-332: "moving MVBS upwards." Please, rephrase because MVBS cannot move anywhere.

Thank you. We changed this sentence in lines 402-403 as follows: "*Downwelling events disrupt the MVBS diurnal signal in the opposite way compared to upwellings and moonlight, moving **acoustic backscatters** upwards*".

30. L337: "It seems that downwelling #3D is entirely dominated by the moon..." Do the authors claim that the Moon impacts downwelling somehow?

We do not claim this, of course. We modify this sentence in line 409 for clarity: "*It seems that **event #3D** is entirely dominated by the moon...*".

31. L376-377: "It appears that this MVBS anomaly is attributed to the eddy-entrained suspended particles commonly recorded in this area (O'Brien et al., 2011)." If the disruptions of diurnal cycle are attributed to higher concentration of suspended materials in the eddy, why do we see such an unusual pattern in vertical velocity around the core in Fig. 7? I mean very high positive anomalies of the vertical velocity during the day time at 68- and 88-m depths. And a more general question: the authors noted in Discussion that "zooplankton likely avoids enhanced water dynamics" (L539). In that regards, why we do not see vertical migration of both signs in layers above and below the eddy core where we have the strongest currents?

For the first part of this comment, please see our response to your comment #37, and text newly introduced in lines 355-363. Responding the second part of this comment, we clarify that below the eddy core the DVM signal was inverted (lines 444-445). Above the eddy, MVBS is slightly elevated during the light time (Figures 7c, 7g and 10c). The possible interpretation is that zooplankton partially occupied the sub-surface layer during the light time avoiding enhanced water dynamics attributed to eddy passing. To present our results on eddy impact on DVM in a more clear way, we modified our sentence in lines 445-446 as follows: "*At 28-48 m depth, however, MVBS was not **significantly** modified. **Nevertheless, MVBS was slightly elevated during the light time (Figures 7c, 7g and 10c)***".

32. L378: "December/January 2002/2003 ". Check the period.

Changed to "***2003/2004***" in lines 450 and 456, thank you.

33. L403-404: "Our results show that DVM responds to (i) seasonality of the sunlight, (ii) moonlight, and (iii) seasonality of sea-ice cover that attenuates light transmission to the water column." I intuitively

agree with that statement, but I should emphasize that the authors partially fail to demonstrate DVM associated with sea ice changes and moonlight variations. See my other comments.

Please see our responses to your comments #2, 4, 12, 14, 20, 21, 22, 27, 28, and 34.

34. L445-446: "The inter-annual variability in estimated under-ice illuminance is entirely attributed to the sea-ice thickness." How was this conclusion made if the mean annual cycle for sea ice thickness was used to simulate the under-ice illuminance? In Fig. 3, for example, I see no difference in ice draft among 2003/04 and 2004/05.

Following general comment #2 by Reviewer #2, we completely removed a mean seasonal cycle of ice thickness by Melling et al. (2005) and Krishfield et al. (2014) from our simulations of under ice illuminance. Following the recommendations provided by Reviewer #2, we use PIOMAS and HYCOM + CICE simulated sea-ice thickness data derived for the mooring position. Panel b in Figure 3 now shows PIOMAS-and HYCOM + CICE derived sea-ice thickness data used for computing under-ice illumination in Figures 3d and 3e, respectively. Figures 7b and 8b were recomputed for HYCOM + CICE data on sea-ice thickness.

35. L446-448: "During the ice season, the mean cloud cover (~40%) showed insignificant interannual variability (Figure 6a); thus, the cloud cover was not taken into account." This, again, raises the question about what was included in the model to simulate light illuminance.

Description of the model to simulate light illuminance in lines 161-172 was improved following comment #12 by Reviewer #2. We specifically make note that "***Cloud cover information was not utilized by this model due to high uncertainty of the cloud cover data (Liu and Key, 2016)***", lines 171-172.

36. L550-551: "It appears that upwelling, downwelling, and eddies disrupt DVM by generating a water layer with an enhanced gradient of horizontal velocity." Could the author explain how "the wind-driven barotropic flow generated by upwelling and downwelling wind forcing" (L543) can enhance velocity gradients? If barotropic means depth-uniform current, it cannot produce vertical gradients. Or do the authors mean lateral gradients?

The wind-driven barotropic flow generated by upwelling and downwelling is superimposed on the background bottom-intensified shelfbreak current. For downwelling, this flow amplifies the depth-intensified background baroclinic circulation with enhanced Pacific water transport towards the Canadian Arctic Archipelago. For upwelling, the shelfbreak current is reversed, which results in surface-intensified flow in the opposite direction (Dmitrenko et al., 2018). This is schematically depicted in Figure 11c and explained in figure caption in lines 1041-1044: *"(c) Schematic depiction suggesting generation of the surface-intensified (blue curve) and depth-intensified (red curve) along-slope currents as a result of upwelling and downwelling, respectively, superimposed on the hypothetical bottom-intensified shelfbreak current (black dashed curve) following Dmitrenko et al. (2018)"*. The impact of velocity lateral gradients (cross-slope transport), comprised of upwelling and downwelling, is discussed in lines 634-641: *"The characteristic feature of high-energetic events recorded at CA13 is the depth-dependent behavior of the horizontal flow. For upwelling and downwelling over the eastern Beaufort Sea continental slope, this feature is generated by the superposition of the background and wind-forced flow (Dmitrenko et al., 2018). The wind-driven barotropic flow generated by upwelling and downwelling wind forcing is superimposed on the background bottom-intensified shelfbreak current depicted by a dashed*

*line in Figure 11c (Dmitrenko et al., 2018). For the downwelling storms, this effect amplifies the depth-intensified background circulation with enhanced PWW transport towards the Canadian Arctic Archipelago (Figure 11b and 11c, right). For the upwelling storms, the shelfbreak current is reversed, which results in surface-intensified flow moving in the opposite direction (Dmitrenko et al., 2018 and Figures 11a and 11c, left)"*. The impact of along-slope transport, comprised by upwelling and downwelling, is discussed below in lines 662-665: *"We speculate that DVM disruptions attributed to upwelling and downwelling are primarily dominated by along-slope transport rather than the cross-slope transport. In addition to enhancing the cross-slope transport, upwelling and downwelling over the Beaufort Sea continental slope strongly modify along-slope transport through generating depth-dependent currents over the continental slope (Figure 11; Dmitrenko et al., 2016, 2018)"*.

37. L594 and Fig. 7: This paragraph is confusing. Why do the events of upwelling and downwelling (e.g., 9U and 10D, and many others) have the same positive sign for the vertical velocity within the entire water layer? If we assume that the response of zooplankton to those events is to avoid layers with enhanced water dynamics (l.601), we should see the opposite direction in zooplankton migration in the case of bottom-intensified downwelling and surface-intensified upwelling. However, following Fig.7, this is not the case. Keeping in mind my previous comment to L376-377, I would say that the presented materials on vertical velocities do not support the author's conclusion at all.

We disagree with this comment by Reviewer #2. During the upwelling and downwelling events, the vertical velocity is very noisy (Figures 8c-8g). The vertical velocity component is much more sensitive than the horizontal component to biases inherent in the measurement process itself, such as spatial inhomogeneity of the flow field and errors in the ADCP tilt angle, owing to the fact that the vertical velocity component is typically an order of magnitude smaller – Ott (2005, doi: 10.1016/j.csr.2004.09.007). Ott (2005) also pointed out that the spatial inhomogeneity of the flow can introduce errors in the measured vertical velocity component. Small errors in the measured pitch, roll and tilt can also lead to significant contamination of the vertical velocity. This is consistent with contamination of the vertical velocity observed during high energetic events at CA13 such as upwellings, downwellings and eddy passing.  Thus, vertical velocity cannot be used for interpretation of DVM modifications imposed by upwellings, downwellings, and eddies. Therefore, we build our conclusions based on vertical redistribution of the acoustic backscatter (Figures 7c-7g, 9a and 9b). To clarify our approach, we introduced a new paragraph in lines 355-363 as follows: *"**The vertical velocity component is very sensitive to spatial inhomogeneity of the flow field and errors in the ADCP tilt angle, introducing errors and significant contamination to the measured vertical velocity component (Ott, 2005). This is consistent with contamination of the vertical velocity data observed during upwellings, downwellings and eddy passing (Figures 8c-8g). Deviations of the vertical velocity diurnal pattern can also be attributed to a more dynamical (turbulent) state of the environment associated with high-velocity currents.** In what follows, we are only interested in the vertical velocity estimates, which are sensitive to the MVBS diurnal cycling. **The contaminated vertical velocity data cannot be used for interpretation of DVM modifications imposed by upwellings, downwellings, and eddies. Therefore, our analysis on impact of the major energetic events on DVM is entirely based on vertical redistribution of the acoustic backscatters (Figures 7c-7g, 9a and 9b)**"*.

38. L604-611. Please, see my general comment regarding the analysis of zooplankton.

Following comments #1 by Reviewer #2 and #2 and 6 by Reviewer #1, this section was omitted.

**39. L621-622: "The ADCP data are available through the Polar Data Catalogue at https://www.polardata.ca/pdcsearch/, CCIN Reference #11653 (Gratton et al., unpublished)." This dataset is not available using the CCIN Reference provided.**

The ADCP data are available using this CCIN Reference number. Please click on DATA/MAP:

[Figure]

This will lead you to the data sets:

[Figure]

40. Figure 7: I would suggest the author use a different color scheme for this figure to separate positive and negative vertical velocities in a clearer way. For example, using a white color at zero velocity may help.

We tried different color codes for this figure (Figure 8 in the revised version). However, the light green was found to produce the best possible artificial visual boundary to red and blue, depicting the upward and downward movements, respectively.

---

## Author Response (AR2)

Dear Editor and Reviewer #2,

Following final comment by Reviewer #2 (below), we introduced description of the performed filtration for the vertical velocity component, as requested by Reviewer #2.

**Reviewer #2:** The authors are mute on specific details on how to separate good and bad data in this situation. Without solid proof, this approach does not seem to be appropriate. Therefore, I suggest filtering out suspicious data from Fig. 8, showing only those velocities for which the authors are sure of their quality. Some modifications of the Data section (Section 2) describing the performed filtration would also be helpful.

Following this comment, we:
- (i) Omitted text in lines 360-368;
- (ii) Introduced new text in section 3 (Methods) in lines 149-161: "*Using vertical velocity for DVM interpretation is not intuitive. The vertical velocity component is very sensitive to spatial inhomogeneity of the flow field and errors in the ADCP tilt angle, introducing errors and significant contamination to the measured vertical velocity component (Ott, 2005). Deviations of the vertical velocity diurnal pattern can also be attributed to a more dynamical (turbulent) state of the environment associated with high-velocity currents. In what follows, we are only interested in the vertical velocity estimates, which are sensitive to the MVBS diurnal cycling. For this analysis, the vertical velocity time series were filtered as following. We removed diurnal cycling and low frequency variability using 24-hour and 90-day running mean, respectively. All velocity values exceeding one standard deviation of the mean for the residual time series are considered as noise attributed to spatial inhomogeneity of the flow field and errors in the ADCP tilt angle. In what follows, we show that the contaminated vertical velocity data assigned to upwellings, downwellings, and eddies. Thus, they cannot be used for interpretation of DVM modifications imposed by these major high-velocity events. Therefore, our analysis on impact of the major energetic events on DVM is entirely based on vertical redistribution of the acoustic backscatters*";
- (iii) We modified Figure 8 depicting time periods when vertical velocity data are considered to be contaminated (at the top of Figure 8c);

[revised manuscript text omitted]